# Modality Matters: Universal Time Series Modeling via Channel Dependency Search

## Abstract

The expanding development of wireless and mobile devices results in a proliferation of multivariate time series data, enabling various analytical tasks, e.g., forecasting, classification, and anomaly detection. Most existing time series modeling methods are dedicated to developing task-specific models due to the heterogeneous dimensionalities, resulting in inefficient resource utilization and limited cross-domain transferability. To address this issue, this study achieves a unified paradigm transcending task boundaries and proposes a universal modality-aware Time series modeling framework leveraging Channel Dependency Search named TimeCDS. Specifically, TimeCDS innovatively identifies a certain number of representative features by projecting the heterogeneous time series features into the hierarchical spaces and dynamically modeling their inter-channel relationships to alleviate the heterogeneity issue. A novel time series imaging method is then proposed to automatically introduce the image modality from sequences, facilitating the comprehensive temporal-spatial pattern extraction. Further, a dual-branch architecture is designed to process the sequential data and the visual representations simultaneously, exploiting the complementary cross-modal features through the proposed Cross-Modal Attention and Dynamic Weighted-Averaging. Extensive experiments across different analytical tasks demonstrate the consistently superior performance of TimeCDS, outperforming existing state-of-the-art baselines by up to 15.9%. The code of TimeCDS is publicly available at `https://anonymous.4open.science/r/TimeCDS/`.

## 1 Introduction

The widespread deployment of edge devices generates massive volumes of time series data, enabling various analytical tasks (Hettige et al., 2024; Jiang et al., 2025), e.g., forecasting (Qiu et al., 2024), classification (Campos et al., 2023), and anomaly detection (Liu et al., 2024b). Effective time series analytics facilitates a range of real-world applications (Liu et al., 2024g; Shao et al., 2025), such as traffic prediction (Li et al., 2023; Yi et al., 2024) and fraud detection (Bolton & Hand, 2002) Despite these remarkable advances, most contemporary time series methodologies face some fundamental limitations when confronted with the heterogeneous nature of real-world applications (Liu et al., 2024c;d). Current approaches predominantly operate under the assumption of homogeneous data structures and consistent dimensionalities (Liu et al., 2024g), which severely constrains their applicability in cross-task scenarios where temporal sequences exhibit vastly different variable counts, sampling frequencies, and semantic meanings. This dimensional heterogeneity creates a critical bottleneck that prevents the development of truly universal time series models capable of leveraging knowledge across diverse domains (Xu et al., 2022).

Recent efforts have attempted to address these issues (Liu et al., 2025b; 2024b). For example, large language model-based approaches (Liu et al., 2025b; 2024d) have explored the transformation of time series data into textual representations, enabling the utilization of pre-trained language models for temporal reasoning. Concurrently, image-based methodologies have investigated the conversion of time series into visual representations, treating temporal sequences as two-dimensional structures amenable to computer vision techniques (Liu et al., 2024b). Especially, VisionTS demonstrates that appropriate visual representation exhibits typical time series features, e.g., trend, seasonality, and stationarity, facilitating temporal dependency capturing (Mouxiang Chen, 2025). While these approaches demonstrate promising results in specific contexts, they have the following limitations.

Purely relying on textual representations may lose crucial temporal granularity and numerical precision, while image-based methods often struggle to preserve the sequential nature and inter-variable relationships that are fundamentally important to time series understanding (Campos et al., 2023).

In this paper, we consider to explicitly combine temporal sequence modeling with spatial image-based representations through multimodal fusion mechanisms. The explicit multimodal integration is expected to help capture more comprehensive temporal-spatial information for more effective time series analysis. To this end, we need to address the following challenges.

*C1. How to effectively align the multi-dimensional time series across heterogeneous tasks?* Current approaches either concentrate solely on fixed-dimensional time series data or rely on domain-specific preprocessing pipelines through rigid architectural constraints (Rui et al., 2024). Such inflexible designs prevent them from processing time series with vastly different variable counts and semantic meanings that could provide significant cross-domain knowledge transfer. Further, the dimensional constraints embedded in existing architectures are particularly limiting for cross-domain deployment scenarios (Liang & Wang, 2024). Effectively aligning multi-dimensional time series from heterogeneous domains while preserving their intrinsic characteristics and enabling knowledge transfer are the cornerstones to solving the cross-domain time series modeling problem.

*C2. How to capture both the temporal dynamics and spatial correlations of time series?* Existing approaches face difficulties in simultaneously capturing the sequential temporal evolution and the inter-variable spatial relationships due to the architectural limitations (Liu et al., 2025a). Thus they cannot fully exploit the rich temporal-spatial information inherent in time series data. Naive integration of the temporal and spatial modalities may lead to conflicting optimization objectives, blurring the distinction between sequential dynamics and instantaneous correlations (Mouatadid et al., 2024). Therefore, it is challenging to effectively decompose and model both temporal periodicity patterns and spatial relational structures, ensuring comprehensive representation learning.

*C3. How to effectively fuse complementary cross-modal representations?* Current fusion strategies struggle with balancing the contributions from different modalities due to the static weighting schemes and insufficient cross-modal interaction mechanisms (Cheng et al., 2024). This limits the model's ability to adaptively leverage the strengths of each representation modality. Excessive reliance on one modality may lead to suboptimal performance, while improper fusion may introduce noise and conflicting signals that degrade the overall model effectiveness (Ekambaram et al., 2023). Therefore, developing adaptive fusion mechanisms that can dynamically integrate temporal sequence features with spatial image-based representations based on input characteristics and task requirements remains a critical challenge.

To address these challenges, we propose a novel modality-aware framework that synergistically combines temporal sequence encoding with spatial image-based representations through sophisticated cross-modal attention mechanisms. Our approach tackles the dimensional alignment problem through an independent similarity search strategy that enables effective processing of multi-variable time series regardless of their original dimensionality. Rather than forcing all inputs into a fixed architectural template, we develop adaptive mechanisms that preserve the intrinsic characteristics of each domain while enabling knowledge transfer across heterogeneous data sources through dimensionality-agnostic feature extraction and alignment procedures. For effective temporal-spatial modeling, we introduce a dual-branch encoding architecture, where the time series encoding branch captures sequential dynamics through patch-based Transformer encoding, while the spatial branch models inter-variable relationships through image-like convolutions that treat reshaped time series as spatial structures. The time image encoding branch decomposes time series modeling into complementary perspectives: periodicity extraction, relational matrix modeling, and phase-amplitude analysis, based on time series imaging. This decomposition reflects the fundamental mathematical properties of time series data—periodicity captures the cyclical patterns inherent in temporal phenomena, relational matrices encode the correlation structures among variables, and phase-amplitude analysis preserves the frequency domain characteristics crucial for understanding temporal dynamics. The cross-modal fusion challenge is addressed through the proposed Cross-Modal Attention Mechanism (CMAM) combined with Dynamic Weighted-Averaging Mechanism (DWAM), which adaptively determines the optimal integration strategy based on the specific characteristics of each input sequence. Rather than static weight fusion or simple concatenation, these mechanisms enable dynamic interaction between temporal and spatial representations, ensuring that the final fused rep-

resentation optimally balances the contributions from each modality based on their relevance to the specific task and input characteristics while reducing redundancy and conflicting signals.

The major contributions are summarized as follows:

- We introduce a unified cross-modal architecture that effectively addresses the dimensional heterogeneity problem in universal time series modeling through independent similarity search and adaptive dimensionality alignment mechanisms.
- We propose a novel dual-branch encoding strategy combined with sophisticated cross-modal attention mechanisms that optimally integrate temporal sequence dynamics with spatial structural representations, enabling superior modeling of complex time series patterns.
- Extensive experiments are conducted on real-world datasets, proving the effectiveness of the proposed TimeCDS for universal time series analytics, including prediction, classification, and anomaly detection, achieving a comprehensive surpass over the SOTA.

## 2 RELATED WORK

**Task-specific Time Series Modeling.** With the growing availability of time series data and the resulting rich downstream applications, time series modeling has attracted increasing interest in both academia and industry (Qiu et al., 2024; Liu et al., 2024b; Campos et al., 2023; Rui et al., 2024). Traditional time series modeling methods are mostly task-specific, which are developed for specific time series tasks, such as forecasting (Qiu et al., 2024; Han et al., 2024; Liu et al., 2025a), classification (Campos et al., 2023; Liang & Wang, 2024), and anomaly detection (Liu et al., 2024b; Chen et al., 2023; Schmidl et al., 2025). In the early stage, statistics-based time series modeling methods became mainstream, such as ARIMA (Shekhar & Williams, 2007). Numerous neural architectures have been developed for effective task-specific time series modeling, including Temporal Convolutional Networks (Cheng et al., 2024), Recurrent Neural Networks (Liu et al., 2021), Multilayer Perceptrons (Ekambaram et al., 2023), and Transformers (Liu et al., 2024f; 2025a; Chen et al., 2023). However, these methods are mainly invented for specific tasks, falling short in handling different tasks simultaneously.

**Universal Time Series Modeling.** Universal time series modeling methods often develop a universal modeling paradigm that handles different tasks (Liu et al., 2024c;d), aiming to overcome the limitations of traditional task-specific models by means of large-scale pretraining (Manuso et al., 2021) and adaptive fine-tuning (Nguyen et al., 2024). These models often adopt a unified architecture that supports a range of time series related tasks, including forecasting (Nie et al., 2023), anomaly detection (Gao et al., 2024), and classification (Nguyen et al., 2024). The core idea is to project time series from different tasks into a general feature space to understand their common temporal semantics. Recent studies have sought to develop novel architectures to model diverse time series (Wu et al., 2023; Gao et al., 2024). However, LLM-based methods require significant computational resources, resulting in high training costs. To be specific, these methods process each channel of the time series individually. However, the correlations across channels are ignored, which may result in significant model performance deterioration.

## 3 METHODOLOGY

We proceed to detail the proposed universal time series modeling framework, TimeCDS. As shown in the figure 1, TimeCDS consists of three major components: (1) Input projection, (2) Dual-branch Encoding, and (3) Cross-Modal Alignment. We then provide specifics on each module in the framework.

### 3.1 INPUT PROJECTION

In this paper, we defined the cross-task datasets as $\mathcal{D} = \{D_1, D_2, \ldots, D_N\}$, where each domain $D_i$ contains multiple time series $D_i = \{T_1^i, T_2^i, \ldots, T_{n_i}^i\}$. Most existing studies Liu et al. (2024d) employ channel-independence mechanisms to accommodate cross-task time series with different dimensionalities, which, however, may ignore the correlations across channels. In contrast, we adopt channel mixing to capture cross-variable interactions, which may facilitate comprehensive temporal

Figure 1: TimeCDS Framework Overview

pattern extraction. In the dimensionality reduction and fusion process of multivariate time series, we propose a distance-weighted representative feature selection mechanism via channel dependency search (see Figure 2). Initially, the original $N$-dimensional multivariate time series is decomposed into $N$ one-dimensional time series via channel-wise independent processing. To achieve the target of fixed-dimensionality reduction, a subset of $K < N$ most representative time series channels is selected as core features. This selection aims to retain the subset with the maximal information content and expressive power among the multivariate data. The selection criterion can be based on centrality measures derived from the Hierarchical Navigable Small World (HNSW) graph (Malkov & Yashunin, 2018), which can be formalized as:

$$\mathcal{S} = \arg \max_{\mathcal{S} \subseteq \{1,...,N\}, |\mathcal{S}|=K} \sum_{i \in \mathcal{S}} \mathcal{R}(i), \tag{1}$$

where $\mathcal{R}(i)$ denotes the representativeness score of channel $i$.

**Channel Dependency Search.** For the remaining $(N - K)$ discarded channels, their information is integrated via a distance-weighted fusion mechanism. Specifically, each discarded channel's time series is mapped onto the closest representative channels weighted inversely by their distance:

$$\mathbf{x}'_j = \sum_{i \in \mathcal{S}} w_{ji} \mathbf{x}_i, \quad w_{ji} = \frac{\exp(-d(\mathbf{x}_j, \mathbf{x}_i))}{\sum_{i' \in \mathcal{S}} \exp(-d(\mathbf{x}_j, \mathbf{x}_{i'}))}, \tag{2}$$

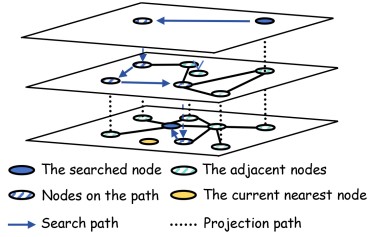

where $\mathbf{x}_j$ is the time series of the discarded channel $j$, $d(\cdot, \cdot)$ is a distance metric between time series, and $\mathbf{x}'_j$ is the fused representation obtained by weighted aggregation. This weighted fusion ensures that the discarded features' information is not completely lost, but softly integrated into the low-dimensional feature space via their nearest representative channels.

Through the above procedure, a fixed dimension $K$ representation of the multivariate time series is obtained, preserving the majority of the original variables' infor-

Figure 2: Channel Dependency Search

mation and structural characteristics. Leveraging this HNSW structure for nearest neighbor search on this reduced space further ensures efficient and effective similarity queries.

### 3.2 DUAL-BRANCH ENCODING

Based on the multi-variate time series dimensionality reduction and fusion achieved by HNSW, a fixed-dimension $K$-dimensional reconstructed multi-variable time series representation $\mathbf{X} \in \mathbb{R}^{B \times K \times T}$ is obtained, where $B$ denotes the batch size and $T$ denotes the sequence length. To further enhance the representational power and discriminative capability of this fused representation, a multimodal encoding module is designed to deeply extract features from the reconstructed $K$-dimensional time series. This module consists of two parallel branches, focusing respectively on the temporal dynamics of the sequence itself and the spatial structural features from its image-like representation.

**Time Series Encoding Branch.** The time series encoding branch takes the fused and reconstructed $K$-dimensional time series $\mathbf{X} \in \mathbb{R}^{B \times K \times T}$ as input. To effectively capture local temporal dynamics,

the sequence is first divided along the time axis into $M$ equal-length patches, each of length $L$, satisfying $T = M \times L$:

$$\mathbf{P}_m = \mathbf{X}_{[:,:,(m-1)L+1:mL]} \in \mathbb{R}^{B \times K \times L}, \quad m = 1, \dots, M \tag{3}$$

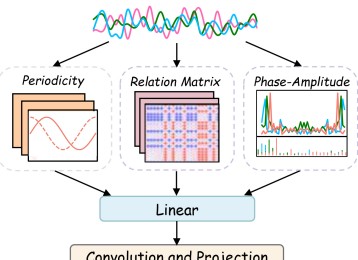

Figure 3: Time Series Imaging

Each patch represents a localized temporal segment of the multi-variable sequence, facilitating the modeling of fine-grained temporal variations. Subsequently, each patch is encoded by a Transformer encoder based on the self-attention mechanism. The Transformer encoder, utilizing multi-head self-attention and feedforward networks, effectively models both local and global temporal dependencies. The encoding process is formally expressed as:

$$\mathbf{Z}_m = \text{TransformerEncoder}(\mathbf{P}_m) \tag{4}$$

By aggregating the encoded representations of all patches via methods such as concatenation or weighted fusion, the comprehensive deep temporal representation is obtained:

$$\mathbf{Z}_{TS} = \text{Aggregation}(\{\mathbf{Z}_m\}_{m=1}^M) \in \mathbb{R}^{B \times K \times d} \tag{5}$$

This branch thoroughly extracts the dynamic temporal features inherent in the HNSW-fused reconstructed time series, thereby enhancing the model's ability to capture complex sequential patterns.

**Time Image Encoding Branch.** To complement the temporal encoding with spatial structural understanding among variables, the input sequence $\mathbf{X}$ is reshaped into a four-dimensional tensor appropriate for image-like feature extraction via a novel time series imaging method, as shown in Figure 3: $\mathbf{X} \to \mathbf{I} \in \mathbb{R}^{B \times L \times K \times C}$, where $L$ denotes the number of temporal patches, $K$ is the variable dimension, and $C$ stands for the number of feature channels. The imaging process involves three steps as follows:

1. Temporal patch division:

$$\mathbf{P}_l = \mathbf{X}_{[:,:,(l-1)(T/L)+1:l(T/L)]} \in \mathbb{R}^{B \times K \times \frac{T}{L}}, \quad l = 1, \dots, L \tag{6}$$

2. Multi-path feature extraction via three parallel streams:

- **Periodicity Encoder**: Extracts periodic patterns within the sequence based on temporal convolutional networks, outputting a tensor of shape $\mathbb{R}^{B \times L \times K \times 2}$.
- **Relation Matrix Computation**: Computes dependency graphs among variables and temporal patches, generating a single-channel spatial relation map $\mathbb{R}^{B \times L \times K \times 1}$ according to (Hssayni et al., 2022).
- **Phase-Amplitude Encoder**: We decompose phase-amplitude information based on (Ni J & A, 2025), outputting a tensor with the same dimensions as the periodicity encoder $\mathbb{R}^{B \times L \times K \times 2}$ by a linear projection.

3. Feature fusion and projection: $\mathbf{Z}_{concat} = \text{Concat}(\mathbf{P}, \mathbf{R}, \mathbf{F}) \in \mathbb{R}^{B \times L \times K \times 5}$

Subsequently, convolutional and projection layers perform spatial abstraction and compression, yielding the final encoded spatial features:

$$\mathbf{Z}_{TI} = \text{ConvProj}(\mathbf{Z}_{concat}) \in \mathbb{R}^{B \times d'} \tag{7}$$

This branch strengthens the expressiveness regarding inter-variable spatial dependencies within the multi-variable time series, complementing the limitations of solely temporal encoding.

### 3.3 CROSS-MODAL ALIGNMENT

**Cross-Modal Attention Mechanism.** Following the extraction of features from the time series encoding and time image encoding branches, the projected modality-specific features are fed into the Cross-Modal Attention Mechanism (CMAM). This mechanism leverages self-attention to dynamically model the interactions between different modalities, effectively emphasizing complementary information and suppressing redundancy. Formally, given query $Q$, key $K$, and value $V$, the attention is computed as:

$$\text{Attention}(Q, K, V) = \text{softmax}\left(\frac{QK^\top}{\sqrt{d}}\right)V, \tag{8}$$

where $Q$ is derived from one modality's projected features (e.g., $\mathbf{F}_{TS}$) and $K, V$ from the other modality's features (e.g., $\mathbf{F}_{TI}$). This cross-attention facilitates complementary feature integration, resulting in modality-specific attention-weighted representations $\mathbf{F}_{TS}^{attn}$ and $\mathbf{F}_{TI}^{attn}$.

**Dynamic Weighted-Averaging Mechanism.** The attentive modality features generated by CMAM are subsequently fused via the Dynamic Weighted-Averaging Mechanism (DWAM). This module adaptively assigns fusion weights based on the input features, enabling flexible integration tailored to each example:

$$\mathbf{F}_{fusion} = \alpha \cdot \mathbf{F}_{TS}^{attn} + (1 - \alpha) \cdot \mathbf{F}_{TI}^{attn}, \tag{9}$$

where the fusion coefficient $\alpha \in [0, 1]$ is dynamically computed as

$$\alpha = f_\theta(\mathbf{F}_{TS}, \mathbf{F}_{TI}). \tag{10}$$

Here, $f_\theta$ is a lightweight multilayer perceptron (MLP) that dynamically estimates the fusion weight $\alpha$ by jointly considering the concatenated features $\mathbf{F}_{TS}$ and $\mathbf{F}_{TI}$. Formally,

$$\alpha = \sigma\big(\mathbf{W}_2 \, \mathrm{ReLU}(\mathbf{W}_1 \, \mathrm{Concat}(\mathbf{F}_{TS}, \mathbf{F}_{TI}) + \mathbf{b}_1) + \mathbf{b}_2\big), \tag{11}$$

where $\sigma$ denotes the sigmoid function, ensuring $\alpha \in [0, 1]$. This mechanism allows adaptive, sample-wise weighting of modalities, enhancing fusion flexibility and expressiveness.

**Multivariate Transformer Decoder.** The fused embedding $\mathbf{F}_{fusion}$ is subsequently fed into a multivariate Transformer decoder specialized for task-specific output generation. Leveraging the decoder's powerful self-attention mechanism, it models complex temporal and inter-variable dependencies embedded in the fused features, enabling unified multi-task inference across diverse cross-domain time series datasets with enhanced accuracy and robustness.

### 3.4 OPTIMIZATION

This work proposes a unified multimodal framework designed to support multiple heterogeneous time series tasks, including prediction, classification, and anomaly detection. Although the framework enables a shared representation and fusion mechanism across tasks and domains, each task's model parameters $\theta_t$ are optimized independently to accommodate their distinct objectives and data distributions.

Formally, for each task $t \in \{\mathrm{pred}, \mathrm{cls}, \mathrm{anom}\}$, given its corresponding dataset $\mathcal{D}_t = \{(\mathbf{X}_i^{(t)}, y_i^{(t)})\}_{i=1}^{N_t}$, the training objective is to minimize the task-specific loss:

$$\theta_t^* = \arg\min_{\theta_t} \frac{1}{N_t} \sum_{i=1}^{N_t} \mathcal{L}^{(t)}\big(f_{\theta_t}(\mathbf{X}_i^{(t)}), y_i^{(t)}\big), \tag{12}$$

where $f_{\theta_t}(\cdot)$ denotes the model inference for task $t$. In cross-domain time-series analysis, a cross-domain dataset is defined as $\mathcal{D} = \{D_1, D_2, \ldots, D_N\}$, where each domain $D_i$ contains multiple time series $D_i = \{T_1^i, T_2^i, \ldots, T_{n_i}^i\}$.

The goal of the pre-trained model $\mathcal{M}_\Theta$ is to learn general cross-domain feature representations. For this purpose, we design the objective function as follows:

$$\mathcal{L}(\Theta) = -\sum_{i=1}^{N} \sum_{T_j^i \in D_i} \log p_\Theta(T_j^i) + \lambda \sum_{i=1}^{N} \sum_{T_j^i \in D_i} \|\mathbf{z}(T_j^i; \Theta) - \mathbf{z}_{\mathrm{avg}}(\mathcal{D}; \Theta)\|_2^2 \tag{13}$$

where $\mathbf{z}(\cdot; \Theta)$ represents the feature vector representation of a time series, $\mathbf{z}_{\mathrm{avg}}$ is the average of all domain features, and $\lambda$ is a weighting coefficient that balances the generation probability and feature consistency. By independently optimizing $\theta_t$ for each task, the framework maintains specialization and high performance tailored to each task's data characteristics and objectives. Meanwhile, the shared multimodal architecture fosters parameter and representation reuse, enabling effective transfer and robustness in cross-domain, multi-task time series applications. Existing time series foundation models (Liu et al., 2024h; Gao et al., 2024) show that pre-training enhances the model performance since more training data is involved, introducing more useful knowledge. Motivated by this, we pre-train TimeCDS with the UTSD-4G dataset (Liu et al., 2024h) and then finetune it for new datasets.

# 4 EXPERIMENTS

In this section, we systematically evaluate the efficacy of TimeCDS across various tasks within the time series domain, complemented by ablation experiments that elucidate the individual contributions of its components to the overall performance (refer to Section 4.5). To rigorously assess the generalizability of the TimeCDS approach, we conducted extensive empirical analyses across three critical time series tasks: Time Series Forecasting (see Table 1), Anomaly Detection (see Table 2), and Classification (see Figure 4).

## 4.1 DATASETS AND EXPERIMENT SETUP

**Datasets.** The experiments are carried out on certain real-world time series datasets. In terms of forecasting, we employ 8 datasets, including ETTh1, ETTh2, ETTm1, ETTm2, ECL, Trafffc, Weather (Wu et al., 2021), and Solar (Liu et al., 2024f). For anomaly detection, we adopt 5 datasets, including SMD, PSM, SWaT, MSL, and SMAP (Liu et al., 2024a). For time series classification, we employ UEA (Bagnall et al., 2018). Dataset details can be found in Appendix A.

**Baselines.** We compare TimeCDS with the following existing baselines, including 13 time series forecasting baselines, i.e., ARIMA(Stellwagen & Tashman, 2013), DLinear (A Zeng, 2021) TimesNet (Wu et al., 2023), PatchTST (Nie et al., 2023), N-HiTS (Cristian Challu, 2023), iTransformer (Liu et al., 2024e), TimeVLM (Siru Zhong, 2025), TimeLLM (**?**), AutoTimes (Liu et al., 2024g), UniTime (Liu et al., 2024d), UniTS (Gao et al., 2024), Timer (Liu et al., 2024h), and TimerXL (Liu et al., 2025c). For anomaly detection, we compare TimeCDS with ARIMA(Stellwagen & Tashman, 2013), FEDformer (Tian Zhou, 2022), Informer (Haoyi Zhou, 2021), DCdetector (Yiyuan Yang, 2023), Autoformer (Wu et al., 2021), DLinear (A Zeng, 2021), TimesNet (Wu et al., 2023), Series2graph (Ser2graph) (Paul Boniol, 2020), TranAD (Shreshth Tuli, 2022), IMDIFFUSION (Chen et al., 2023) and Timer (Liu et al., 2024h). For time series classification, 13 baselines are selected, i.e., LSTM (Shi et al., 2015), LSTNET (G Lai, 2018), Informer (Haoyi Zhou, 2021), FEDformer (Tian Zhou, 2022), Full Attention (Attn) (Haoqing Wang, 2023), Rocket (Dempster et al., 2020), InceptionTime (Ismail Fawaz et al., 2020), TCN (Bai et al., 2018), LIGHTTS (Campos et al., 2023), DLinear (A Zeng, 2021), TimesNet(Wu et al., 2023), UniTS (Gao et al., 2024), and Timer (Liu et al., 2024h). Please note that TimeLLM is a large language mode based time series analytics method, while Timer, TimerXL, and UniTS are time series foundation models. We follow the default hyperparameter setting in the original paper or the associated code of baselines, enabling fair comparison. The implementation details are given in Appendix B.

**Evaluation Metrics.** We adopt mean squared error (MSE) and mean absolute error (MAE) as evaluation metrics for time series forecasting (Qiu et al., 2024). The F1-score (F1), AUC-ROC, and PATE (Ghorbani et al., 2024) are adopted as main evaluation metrics for time series anomaly detection. Additionally, F1 and accuracy are used to evaluate time series classification. More results on more evaluation metrics and model efficiency analysis can be seen in Appendix C.

## 4.2 TIME SERIES FORECASTING

Table 1: Overall Performance Comparison of Time Series Forecasting (Average)

| Models | ETTm1 | | ETTm2 | | ETTh1 | | ETTh2 | | ECL | | Weather | | Traffic | | Solar | |
|---|---|---|---|---|---|---|---|---|---|---|---|---|---|---|---|---|
| | MSE | MAE | MSE | MAE | MSE | MAE | MSE | MAE | MSE | MAE | MSE | MAE | MSE | MAE | MSE | MAE |
| ARIMA(2013) | 1.172 | 0.813 | 2.425 | 1.208 | 1.228 | 0.851 | 3.126 | 1.382 | 0.589 | 0.579 | 0.474 | 0.484 | 1.041 | 0.572 | 1.293 | 1.375 |
| N-HiTS(2023) | 0.452 | 0.461 | 0.305 | 0.370 | 0.493 | 0.514 | 0.436 | 0.470 | 0.210 | 0.313 | 0.279 | 0.317 | 0.457 | 0.344 | 0.285 | 0.307 |
| TimesNet(2023) | 0.525 | 0.521 | 0.411 | 0.453 | 0.578 | 0.570 | 0.527 | 0.547 | 0.231 | 0.333 | 0.294 | 0.323 | 0.632 | 0.352 | 0.243 | 0.325 |
| PatchTST(2023) | 0.439 | 0.460 | 0.361 | 0.411 | 0.480 | 0.502 | 0.393 | 0.405 | 0.199 | 0.298 | 0.257 | 0.298 | 0.421 | 0.305 | 0.232 | 0.299 |
| DLinear (2021) | 0.453 | 0.467 | 0.461 | 0.477 | 0.473 | 0.479 | 0.376 | 0.432 | 0.195 | 0.295 | 0.270 | 0.321 | 0.453 | 0.328 | 0.252 | 0.313 |
| UniTime(2024d) | 0.419 | 0.626 | 0.470 | 0.494 | 0.634 | 0.623 | 0.555 | 0.563 | 0.189 | 0.429 | 0.265 | 0.489 | 0.404 | 0.397 | 0.227 | 0.492 |
| TimeVLM(2025) | 0.371 | 0.410 | 0.289 | 0.355 | 0.440 | 0.451 | 0.364 | 0.410 | 0.190 | 0.294 | 0.257 | 0.296 | 0.463 | 0.341 | 0.247 | 0.297 |
| iTransformer(2024e) | 0.428 | 0.436 | 0.316 | 0.358 | 0.466 | 0.487 | 0.412 | 0.445 | 0.197 | 0.286 | 0.289 | 0.306 | 0.460 | 0.305 | 0.269 | 0.290 |
| UniTS (2024) | 0.459 | 0.469 | 0.478 | 0.494 | 0.474 | _0.426_ | 0.384 | **0.379** | 0.214 | 0.312 | 0.258 | 0.298 | 0.491 | 0.356 | 0.266 | 0.333 |
| AutoTimes (2024g) | 0.438 | 0.452 | 0.453 | 0.511 | 0.617 | 0.640 | 0.538 | 0.580 | 0.355 | _0.283_ | 0.491 | 0.303 | 0.614 | 0.369 | 0.422 | _0.272_ |
| TimeLLM (2024) | 0.442 | 0.467 | 0.493 | 0.526 | 0.657 | 0.655 | 0.578 | 0.595 | 0.211 | 0.318 | _0.255_ | _0.295_ | 0.440 | 0.333 | 0.293 | 0.365 |
| Timer(2024h) | 0.384 | 0.418 | 0.295 | 0.354 | _0.420_ | 0.448 | 0.370 | 0.417 | 0.199 | 0.295 | 0.263 | 0.301 | **0.361** | **0.268** | 0.352 | 0.422 |
| TimerXL (2025c) | _0.366_ | _0.407_ | _0.288_ | _0.350_ | 0.441 | 0.464 | _0.363_ | 0.417 | 0.199 | 0.295 | 0.264 | 0.300 | _0.393_ | _0.296_ | _0.223_ | 0.295 |
| TimeCDS | **0.356** | **0.393** | **0.258** | **0.320** | **0.395** | 0.419 | **0.340** | _0.390_ | **0.182** | **0.262** | **0.242** | **0.284** | 0.431 | 0.337 | **0.192** | **0.203** |

Time series forecasting is a central task in time series analysis. To evaluate the performance of TimeCDS in time series forecasting, we compared it with 13 baseline models on standard benchmarks, including ETTh, ECL, Weather, Traffic, and Solar. We train and test the time series foundations models, i.e., TimeCDS, Timer, TimeXL, and UniTS, on all datasets simultaneously. As shown

in Table 1, we report the average performance of forecasting, he complete results can be found in Table 4. TimeCDS achieves the best performance in most cases across various prediction lengths from 96 to 720. TimeCDS performs better than the best among the baselines by up to 15.9%. We see that ARIMA has the worst performance. This is because the traditional statistics-based method is often shallow, failing to capture the complex temporal correlations. In addition, time series foundation models perform better than transformer-based methods, e.g., PatchTST, and MLP-based methods, e.g., TimesNet, in most cases, showing their superior generalization capabilities, enhancing model performance.

## 4.3 ANOMALY DETECTION

Table 2: Overall Performance Comparison of Time Series Anomaly Detection ($\times 100\%$)

| Models | SMD | | | PSM | | | SWaT | | | MSL | | | SMAP | | | Average | | |
|---|---|---|---|---|---|---|---|---|---|---|---|---|---|---|---|---|---|---|
| | F1 | AUC | PATE | F1 | AUC | PATE | F1 | AUC | PATE | F1 | AUC | PATE | F1 | AUC | PATE | F1 | AUC | PATE |
| ARIMA(2013) | 31.74 | 26.95 | 13.59 | 36.21 | 31.25 | 42.13 | 32.99 | 34.98 | 11.59 | 26.95 | 29.99 | 16.39 | 24.69 | 27.10 | 16.98 | 30.52 | 30.05 | 20.14 |
| FEDformer(2022) | 75.63 | 68.32 | 57.90 | 70.31 | 78.45 | 55.34 | 52.36 | 59.48 | 43.63 | 48.69 | 50.98 | 65.32 | 51.46 | 49.47 | 48.01 | 59.69 | 65.34 | 54.04 |
| Informer(2021) | 69.35 | 82.48 | 67.49 | 65.20 | 74.32 | 57.64 | 33.65 | 54.31 | 23.31 | 66.54 | 78.31 | 77.31 | 76.86 | 63.73 | 64.85 | 62.32 | 70.63 | 58.12 |
| DCdetector(2023) | 65.48 | 80.36 | 68.93 | 60.38 | 68.32 | 51.36 | 68.36 | 62.31 | 67.41 | 56.70 | 66.54 | 56.30 | 23.88 | 60.07 | 40.80 | 54.96 | 67.52 | 56.96 |
| Autoformer(2021) | 57.90 | 58.69 | 62.47 | 59.58 | 68.75 | 65.30 | 67.54 | 56.63 | 65.25 | 44.68 | 59.62 | 66.75 | 40.40 | 57.46 | 46.73 | 54.02 | 60.23 | 61.30 |
| DLinear(2021) | 68.49 | 74.75 | 55.46 | 70.65 | 76.49 | 43.59 | 60.70 | 70.53 | 56.78 | 65.31 | 69.49 | 58.61 | 66.75 | 68.84 | 55.16 | 66.38 | 72.02 | 53.92 |
| TimesNet (2023) | 69.74 | 73.85 | 64.90 | 73.21 | 75.65 | 72.60 | 65.19 | 68.93 | 58.36 | 75.36 | 77.69 | 85.36 | 67.15 | 69.98 | 73.83 | 70.13 | 73.22 | 71.01 |
| Ser2graph(2020) | 63.51 | 71.65 | 55.32 | 64.98 | 69.46 | 66.79 | 56.84 | 66.32 | 59.64 | 59.64 | 68.49 | 55.30 | 66.70 | 68.31 | 59.67 | 62.33 | 68.85 | 59.34 |
| TranAD(2022) | 66.31 | 75.65 | 62.98 | 65.39 | 71.85 | 67.59 | 64.25 | 71.36 | 62.39 | 63.59 | 69.75 | 62.98 | 59.62 | 65.31 | 60.81 | 63.83 | 70.78 | 63.35 |
| IMDIFFUSION (2023) | 69.75 | 73.86 | 64.91 | 73.32 | 75.76 | 72.71 | 65.20 | 75.65 | 75.36 | 75.37 | 77.70 | 85.37 | 67.16 | 69.99 | 73.84 | 70.16 | 74.59 | 74.44 |
| Timer(2024h) | 79.65 | 82.36 | 61.87 | 80.02 | 83.54 | 62.39 | 77.06 | 79.65 | 58.67 | 75.09 | 75.63 | 57.31 | 77.63 | 77.17 | 60.81 | 77.89 | 79.67 | 60.21 |
| TimeCDS | 82.61 | 85.96 | 77.31 | 83.26 | 85.64 | 76.59 | 79.67 | 85.95 | 72.69 | 74.62 | 85.45 | 79.56 | 80.06 | 83.55 | 72.50 | 80.52 | 85.31 | 75.73 |

Since the anomalies are usually hidden in the large-scale data, making the data labeling hard, we focus on unsupervised time series anomaly detection, which is to detect the abnormal time points. We evaluate unsupervised point-wise anomaly detection on five benchmarks (SMD, MSL, SMAP, SWaT, PSM) spanning service monitoring, space telemetry, and industrial control. Following Anomaly Transformer, we use fixed-length sliding windows and train via reconstruction. We report F1, AUC, and PATE as our primary metrics as prior works (Ghorbani et al., 2024; Liu et al., 2024a). As shown in Table 2, TimeCDS attains the highest average performance across five benchmarks on F1/AUC/PATE (80.20/85.31/75.73). It ranks first on SMD and PSM across all three metrics; on SMD, TimeCDS reaches 12.4% absolute gain in terms of PATE over the second-best IMDIFFICTION. These results demonstrate that TimeCDS provides stable cross-domain generalization and effective range handling in unsupervised anomaly detection.

## 4.4 CLASSIFICATION

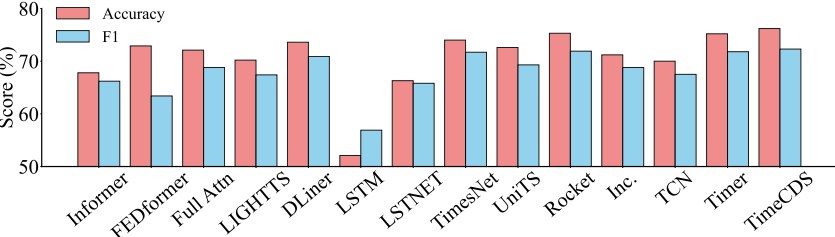

Figure 4: Performance Comparison of classification.

We conduct experiments on time series classifications on 10 subsets of the UEA time series archive. Accuracy and F1-score (F1) are adopted as evaluation metrics. The overall average performance results are provided in Figure 4. TimeCDS performs better than the best among the baselines. Overall, TimeCDS achieves the best results on 10 time series datasets in UEA, which shows that TimeCDS is effective in time series classification. We can see that the time series foundation model Timer achieves the best performance among baselines, showing the promising potential of foundation models. TimeCDS performs better than Timer due to the dual-branch encoding module, which learn an effective representation across two complementary modalities.

## 4.5 ABLATION STUDY

To gain insight into the effects of the different components of TimeCDS, we evaluate three components including 1). *w/o_TS*: TimeCDS without time series encoding branch; 2). *w/o_Image*: TimeCDS without time image encoding branch; 3) *w/o_CMA*: TimeCDS without cross-modal attention mechanism. Figure 5 shows results for forecasting (Figures 5(a) and (b)) and anomaly detection (Figures 5(c) and (d)). Regardless of the datasets, TimeCDS outperforms its counterparts, showing

that these three components are all useful for effective universal time series modeling. TimeCDS obtains MSE and MAE reductions by up to 38.6% and 22.5%, respectively. Further, on all datasets, *w/o_TS* performs worst among all variants. TimeCDS performs better than *w/o_TS* by at least 8%, which indicates the importance of time series encoding branch.

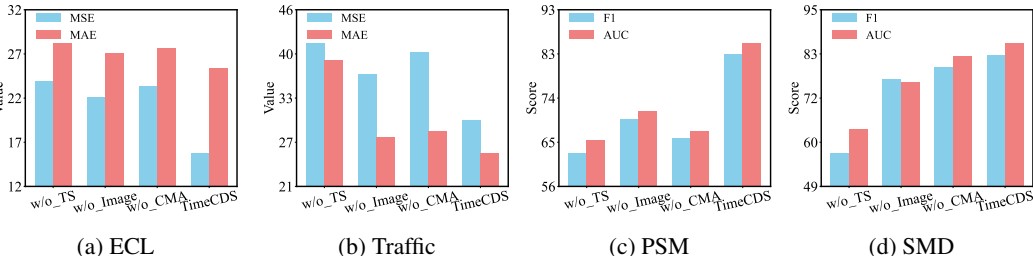

Figure 5: TimeCDS and Its Variants on Four Datasets

## 4.6 CASE STUDY

To intuitively show the effectiveness of the proposed TimeCDS, we provide case studies on ECL and SMD in terms of forecasting and anomaly detection, respectively, as shown in Figure 6. In Figure 6(a), we see that the predictions are highly consistent with the ground truth, demonstrating the effectiveness of TimeCDS. Figure 6(b) shows that TimeCDS successfully identifies the outliers on SMD, demonstrating its superior performance for anomaly detection. These two figures jointly demonstrate that the TimeCDS model can accurately predict future trends and promptly detect anomalies when handling different time series analysis tasks, reflecting its superior capability in universal time series modeling.

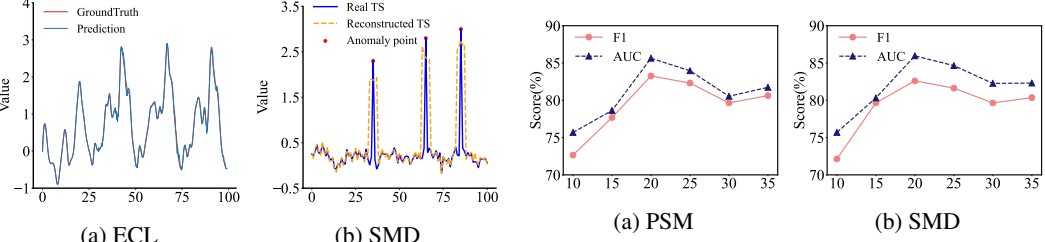

Figure 6: Case Study on Forecasting (ECL) and Anomaly Detection (SMD)

Figure 7: The Effect of $K$.

## 4.7 THE EFFECT OF THE NUMBER OF $K$

We next investigate the effect of the number of $K$ on model performance, which denotes the number of selected representative channels. We vary the value of K=[10, 15, 20, 25, 30, 35]. As shown in Figure 7, we observe that the F1 and AUC curves first increase, then drop, and finally increase slightly. We see that TimeCDS achieves the best performance when $K$ is set to 20, which shows that 20 is the ideal setting in this study. More representative channels may introduce noises, degrading the model performance.

## 5 CONCLUSION

This work presents TimeCDS, a modality-aware dual-branch framework for universal time series forecasting. Extensive experiments on real-world datasets show the effectiveness of the proposed TimeCDS. To accommodate the heterogeneous dimensionalities across domain-varying time series, we propose a channel dependency search strategy to select a certain number of representative channels. To achieve the comprehensive feature extraction, we introduce a time series imaging method and a dual-branch architecture to perform representation learning from sequence and vision simultaneously. A cross-modal alignment module is designed to fuse the complementary cross-modal features. Comprehensive experiments on real datasets offer evidence that TimeCDS achieves the state-of-the-art accuracy. In the future, an interesting research direction is to further improve the pre-training process of TimeCDS with more training data.

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

## A  DATASETS AND METRICS

The table below presents a comprehensive summary of the experimental datasets utilized in the study, categorized into three primary tasks: Forecasting, Anomaly Detection and Classification. Each task is associated with specific benchmark datasets and corresponding metrics that are employed to evaluate the performance of models tailored to these tasks.

Table 3: Summary of Experiment Datasets.

| Tasks | Benchmarks | Metrics |
|---|---|---|
| Forecasting | ETT (4 subsets), ECL, Traffic, Weather, Solar | MSE, MAE |
| Anomaly Detection | SMD, MSL, SWaT, PSM, SMAP | Precision, Recall, F1-Score, AUC, PATE |
| Classification | UEA (10 subsets) | Accuracy, Precision, Recall, F1-Score |

### A.1  FORECASTING

In the context of *Forecasting*, the datasets include ETT, ECL, Traffic, Weather and Solar. These datasets are typically used for time series forecasting tasks, which involve predicting future values at specific time points.

- **ETT.** The ETT dataset includes two hourly-level datasets (ETTh1 and ETTh2) and two 15-minute-level datasets (ETTm1 and ETTm2). Each dataset includes 7 oil and load features of electricity transformers between July 2016 and July 2018.
- **Traffic.** The Traffic dataset contains hourly road occupancy rates obtained from sensors located at San Francisco freeways from 2015 to 2016.
- **Weather.** The Weather dataset contains 21 indicators of weather (e.g., air temperature and humidity), which are collected in Germany. The data is recorded every 10 minutes.
- **ECL.** The ECL dataset captures hourly electricity consumption data from 321 clients.
- **Solar.** The Solar dataset records solar power production from 137 PV plants in 2006, sampled every 10 minutes.

The performance of forecasting models is gauged using the Mean Squared Error (MSE) and the Mean Absolute Error (MAE), which are deffned as follows.

$$\text{MSE} = \frac{1}{n} \sum_{i=1}^{n} (y_i - \hat{y}_i)^2,$$
$$\text{MAE} = \frac{1}{n} \sum_{i=1}^{n} |y_i - \hat{y}_i|, \tag{14}$$

where $n$ is the number of observations, $y_i$ is the actual value of the $i$-th observation, and $\hat{y}_i$ is the predicted value of the $i$-th observation. The MSE measures the average of the squares of the differences between predicted and actual values, with lower values indicating higher accuracy. The MAE, on the other hand, measures the average magnitude of the errors in a set of predictions, without considering their direction, also with lower values indicating better performance.

## A.2 ANOMALY DETECTION

The *Anomaly Detection* task leverages datasets such as SMD , MSL , SWaT , PSM and SMAP. These datasets are employed to identify anomalous patterns or outliers within the data.

- **SMD.** Server Machine Dataset (SMD) is a 5-week-long dataset collected from a large Internet company with 38 feature dimensions.
- **MSL.** Mars Science Laboratory rover (MSL) dataset contains the telemetry anomaly data derived from the incident surprise anomaly reports of spacecraft monitoring systems with 55 feature dimensions.
- **SWaT.** Secure Water Treatment (SWaT) dataset is obtained from 51 sensors of the critical infrastructure system under continuous operations.
- **PSM.** Pooled Server Metrics (PSM) dataset is collected from multiple application servers at eBay with 25 feature dimensions.
- **SMAP.** Soil Moisture Active Passive (SMAP) dataset is a publicly available real-world expert-labeled dataset from NASA. This dataset contains data from 25 entities.

The evaluation metrics for anomaly detection include the F1-Score, which is the harmonic mean of precision and recall, providing a balance between the two especially in cases of class imbalance. The Area Under the Curve (AUC) of the Receiver Operating Characteristic (ROC) curve is also used, with values closer to 1 indicating better model performance. Additionally, We use advanced evaluation metrics of time series anomaly detection: PATE. All metrics are deffined as follows.

$$
\begin{aligned}
\text{Precision} &= \frac{TP}{TP + FP}, \\
F1\text{-Score} &= \frac{2 \times \text{precision} \times \text{recall}}{\text{precision} + \text{recall}}, \\
\text{Recall} &= \frac{TP}{TP + FN}, \\
AUC &= \int_0^1 ROC_{\text{curve}} \, dFPR,
\end{aligned}
\tag{15}
$$

where TP represents True Positive, FP denotes False Positive, and FN is False Negative. FPR (False Positive Rate) represents the proportion of negative instances that are incorrectly classified as positive. AUC represents the Area Under the Receiver Operating Characteristic (ROC) curve.

The Proximity-Aware Time series anomaly Evaluation (PATE) metric assesses model performance through a proximity-aware weighting mechanism. For predefined pre- and post-anomaly buffer sizes $e \in \mathcal{E}$ and $d \in \mathcal{D}$, it computes a weighted Area Under the Precision-Recall curve (AUC-PR). The final score is the average across all buffer combinations:

$$
\text{PATE} = \frac{1}{|\mathcal{E}| \times |\mathcal{D}|} \sum_{e \in \mathcal{E}} \sum_{d \in \mathcal{D}} \text{AUC-PR}_{e,d}
\tag{16}
$$

For each combination $(e, d)$ and threshold $\theta$, the weighted Precision and Recall are derived from time point-level weights $\mathbf{w}^{\text{TP}}(t)$, $\mathbf{w}^{\text{FP}}(t)$, and $\mathbf{w}^{\text{FN}}(t)$, which are assigned based on the spatiotemporal relationship between predictions and ground-truth anomaly segments $\boldsymbol{a}_k = (i_k, n_k)$:

$$
\text{Precision}_{e,d}(\theta) = \frac{\sum_t \mathbf{w}^{\text{TP}}(t)}{\sum_t \mathbf{w}^{\text{TP}}(t) + \sum_t \mathbf{w}^{\text{FP}}(t)}
\tag{17}
$$

$$
\text{Recall}_{e,d}(\theta) = \frac{\sum_t \mathbf{w}^{\text{TP}}(t)}{\sum_t \mathbf{w}^{\text{TP}}(t) + \sum_t \mathbf{w}^{\text{FN}}(t)}
\tag{18}
$$

Key weight assignments include: $\mathbf{w}^{\text{TP}}(t) = 1$ for true detections; $\mathbf{w}^{\text{FP}}(t) = 1$ for points outside any buffer zone; and $\mathbf{w}^{\text{FN}}(t) = 1$ for totally missed anomalies. Weights for points within buffer zones decay with increasing temporal distance from the anomaly segment. Detailed definitions of the weight functions are provided in the original paper (Ghorbani et al., 2024).

### A.3 CLASSIFICATION

Lastly, the *Classification* task utilizes the University of East Anglia (UEA) dataset, which consists of 10 subsets. These datasets are used for classification tasks, where the objective is to assign data points to different categories. The model performance of classification tasks is evaluated using Accuracy, Precision, Recall, and F1-Score. Accuracy is the ratio of correctly classified instances to the total number of instances.

$$\text{Accuracy} = \frac{TP + TN}{TP + TN + FP + FN} \tag{19}$$

### A.4 THE USE OF LARGE LANGUAGE MODELS (LLMS)

LLMs are used in this work solely for auxiliary purposes. Specifically, they assisted in improving the accuracy of writing by identifying and correcting grammatical issues. All research ideas, methodological developments, experiments, and the main body of the manuscript are independently conceived, conducted, and written by the authors.

## B IMPLEMENTATION DETAILS

All experiments are implemented in PyTorch. The pre-trained model is based on the high-quality UTSD-4G dataset and the powerful NVIDIA H200 Tensor Core GPU. The fine-tuned models for downstream tasks and the models for small-scale cross-domain training are deployed on the NVIDIA H20 Tensor Core GPU.

### B.1 PRE-TRAINING

In the pre-training stage, the optimizer we used was AdamW. The attenuation strategy for the learning rate adopted the cosine annealing algorithm. The cosine annealing algorithm is very effective in dynamically adjusting the learning rate. The initial learning rate is set at $10^{-5}$, and the final learning rate is $10^{-6}$. The decay steps are proportional to the number of training steps of 10 epochs. We set the batch size to 2048, which fully utilized the memory bandwidth without encountering out-of-memory errors. Gradient accumulation was therefore unnecessary. The model was trained with mixed precision (FP16/BF16) to accelerate computation while maintaining numerical stability.

### B.2 DOWNSTREAM TASKS

#### B.2.1 FORECASTING FINE-TUNING

The pretrained TimeCDS checkpoint ($K = 20$) was loaded on a single NVIDIA H20. Input series of length 672 were normalized and patched into $M = 7$ segments of size 96, yielding a temporal tensor $\mathbf{X}_{\text{ts}} \in \mathbb{R}^{B \times 20 \times 7}$ and an image tensor $\mathbf{X}_{\text{img}} \in \mathbb{R}^{B \times 7 \times 20 \times 5}$. Training used batch size 64, AdamW ($\beta_1 = 0.9$, $\beta_2 = 0.999$, $\epsilon = 10^{-8}$) and a cosine scheduler with 3-epoch warm-up:

$$\eta_t = \eta_{\min} + (\eta_{\max} - \eta_{\min}) \cdot \frac{1 + \cos(\pi t/T)}{2}, \tag{20}$$

where $\eta_{\max} = 10^{-4}$, $\eta_{\min} = 10^{-5}$, $T = 20$ epochs; early stopping (patience 5) monitored validation MSE. Only LayerNorm, CMAM, DWAM and the decoder were updated.

For inference, **a single model performs rolling forecasting** on each dataset: the same checkpoint is applied with sliding window (stride 1) to iteratively generate 96-step predictions, which are concatenated to produce horizons of $\{96, 192, 336, 720\}$ without retraining or parameter adjustment.

#### B.2.2 ANOMALY DETECTION FINE-TUNING

The same $K = 20$ checkpoint was used on H20. Sliding windows of length 672 (stride 1) generated samples for reconstruction-based training. Batch size was 32 with gradient accumulation 2; optimizer and cosine schedule identical to forecasting. Training ran for 20 epochs with early stopping on validation AUC (patience 5). Backbone weights remained frozen; only CMAM, DWAM and the reconstruction decoder were fine-tuned.

### B.2.3 CLASSIFICATION FINE-TUNING

On H20, variable-length series were padded to the dataset-specific maximum and processed with $K = 20$ channels, producing $\mathbf{X}_{\text{img}} \in \mathbb{R}^{B \times 7 \times 20 \times 5}$. After global average pooling, a linear classifier was appended. Batch size was $128$, label-smoothing $0.01$ was applied, and the cosine LR schedule followed the same parameters as above for 20 epochs, with early stopping on macro-F1 (patience 5). Only LayerNorm, CMAM, DWAM and the classification head were trainable.

### B.3 SMALL-SCALE CROSS-DOMAIN FORECASTING

A single shared backbone was trained on the concatenated ETTh, ECL, Weather, Traffic and Solar. All series were domain-wise standardised, zero-padded to 336 steps, and reduced to K=8 representative channels via the pretrained Channel-Dependency Search. Patch size 48 yields 7 tokens; the imaging branch produces 7×8×5 tensors.

The converged checkpoint was subsequently rolled on each individual dataset to generate horizons {96,192,336,720} without further fine-tuning.

## C ADDITIONAL EXPERIMENTAL RESULTS

### C.1 TIME SERIES FORECASTING

Table 5: Overall performance comparison in forecasting without pre-training

| Models | TimeCDS | | DUET | | UniTS | | TimeLLM | | AutoTimes | | UniTime | | DLinear | | PatchTST | | TimesNet | | ARIMA | |
|---|---|---|---|---|---|---|---|---|---|---|---|---|---|---|---|---|---|---|---|---|
| | MSE | MAE | MSE | MAE | MSE | MAE | MSE | MAE | MSE | MAE | MSE | MAE | MSE | MAE | MSE | MAE | MSE | MAE | MSE | MAE |
| ETTh1 | 0.355 | 0.424 | 0.358 | 0.399 | 0.469 | 0.471 | 0.471 | 0.510 | 0.356 | 0.364 | 0.462 | 0.393 | 0.393 | 0.421 | 0.414 | 0.482 | 0.355 | 0.451 | 1.005 | 0.952 |
| ETTh2 | 0.269 | 0.301 | 0.307 | 0.331 | 0.370 | 0.448 | 0.433 | 0.463 | 0.430 | 0.482 | 0.510 | 0.471 | 0.347 | 0.464 | 0.215 | 0.376 | 0.457 | 0.498 | 0.998 | 0.973 |
| ETTm1 | 0.381 | 0.401 | 0.391 | 0.416 | 0.416 | 0.415 | 0.488 | 0.504 | 0.370 | 0.355 | 0.424 | 0.397 | 0.456 | 0.483 | 0.463 | 0.477 | 0.386 | 0.391 | 1.035 | 0.993 |
| ETTm2 | 0.357 | 0.395 | 0.380 | 0.390 | 0.433 | 0.476 | 0.436 | 0.479 | 0.522 | 0.488 | 0.498 | 0.488 | 0.548 | 0.618 | 0.482 | 0.489 | 0.465 | 0.492 | 1.018 | 1.010 |
| ECL | 0.365 | 0.422 | 0.386 | 0.423 | 0.530 | 0.489 | 0.489 | 0.505 | 0.318 | 0.356 | 0.557 | 0.457 | 0.366 | 0.414 | 0.397 | 0.510 | 0.471 | 0.507 | 1.003 | 0.889 |
| Weather | 0.356 | 0.420 | 0.414 | 0.437 | 0.425 | 0.433 | 0.555 | 0.571 | 0.369 | 0.383 | 0.398 | 0.423 | 0.415 | 0.464 | 0.447 | 0.511 | 0.413 | 0.460 | 1.019 | 1.088 |
| Traffic | 0.492 | 0.415 | 0.524 | 0.476 | 0.440 | 0.489 | 0.557 | 0.533 | 0.505 | 0.501 | 0.466 | 0.502 | 0.414 | 0.454 | 0.390 | 0.458 | 0.439 | 0.460 | 1.070 | 1.038 |
| Solar | 0.207 | 0.233 | 0.213 | 0.240 | 0.268 | 0.258 | 0.234 | 0.262 | 0.202 | 0.225 | 0.253 | 0.188 | 0.200 | 0.217 | 0.192 | 0.208 | 0.239 | 0.243 | 1.061 | 1.067 |

In order to test the cross-domain task capability under a small data scale, our experiment specifically designed a mall-scale cross-domain forecasting experiment. The training of the model was restricted to the datasets of ETTh, ECL, Weather, Traffic and Solar. The joint datasets are divided in [ 6 : 2 : 2 ] ratio, and the performance of the backbone networks of multiple advanced models used for time series prediction was compared. They are DUET, UniTS, TimeLLM, AutoTimes, UniTime, DLinear, PatchTST, Timesnet and ARIMA. From Table 5, in the forecasting task of the mall-scale cross-domain, TimeCDS performed outstandingly, achieving the lowest MSE and MAE values on multiple datasets, demonstrating its generalization ability and prediction accuracy across different datasets. Although slightly inferior to UniTS on the Traffic dataset, TimeCDS still demonstrated good performance. Overall, TimeCDS outperforms or approaches other advanced models on multiple datasets, demonstrating its effectiveness and superiority in handling small-scale cross-domain time series data.

### C.2 TIME SERIES ANOMALY DETCTION

Table 6: Overall performance comparison for time series anomaly detection without pre-training

| Models | SMD | | PSM | | SWaT | | MSL | | SMAP | | AVG | |
|---|---|---|---|---|---|---|---|---|---|---|---|---|
| | Precision | Recall | Precision | Recall | Precision | Recall | Precision | Recall | Precision | Recall | Precision | Recall |
| ARIMA | 42.69 | 31.00 | 35.63 | 29.65 | 29.30 | 34.65 | 26.31 | 31.39 | 21.95 | 25.63 | 31.18 | 30.46 |
| FEDformer | 69.52 | 56.31 | 72.36 | 69.69 | 58.36 | 54.63 | 60.31 | 58.65 | 55.05 | 57.27 | 63.12 | 59.31 |
| Informer | 78.65 | 80.23 | 77.94 | 72.63 | 33.64 | 39.40 | 75.31 | 73.65 | 71.01 | 76.34 | 67.31 | 68.45 |
| DCdetector | 70.31 | 79.65 | 71.35 | 57.36 | 75.21 | 65.72 | 58.36 | 62.34 | 36.32 | 47.03 | 62.31 | 62.42 |
| Autoformer | 55.43 | 53.32 | 62.31 | 63.54 | 55.31 | 52.39 | 52.31 | 49.75 | 35.99 | 44.95 | 52.27 | 52.79 |
| DLinear | 64.51 | 72.36 | 67.21 | 76.54 | 60.49 | 67.54 | 62.37 | 65.89 | 62.02 | 65.17 | 63.32 | 69.50 |
| TimesNet | 78.31 | 72.65 | 71.36 | 74.68 | 63.54 | 65.74 | 72.68 | 82.96 | 70.66 | 83.07 | 71.31 | 75.82 |
| Ser2graph | 75.66 | 72.39 | 69.78 | 73.26 | 59.76 | 59.71 | 62.95 | 66.92 | 53.69 | 62.31 | 64.37 | 66.92 |
| TranAD | 73.52 | 71.32 | 72.31 | 76.51 | 63.44 | 68.54 | 69.58 | 72.31 | 62.31 | 63.14 | 68.23 | 70.36 |
| IMDIFFICTION | 78.32 | 72.66 | 71.37 | 74.79 | 63.65 | 65.85 | 72.69 | 82.97 | 70.67 | 83.08 | 71.34 | 75.87 |
| Timer | 78.65 | 80.21 | 82.14 | 83.65 | 72.63 | 76.99 | 75.65 | 76.59 | 72.03 | 75.46 | 76.22 | 78.58 |
| TimeCDS | 81.65 | 84.31 | 83.56 | 86.59 | 82.46 | 83.62 | 79.65 | 83.32 | 77.78 | 80.36 | 81.02 | 83.64 |

Table 6 provides supplementary performance metrics for each model in the time series anomaly detection task, including Precision and Recall. The TimeCDS model proposed in this study demonstrated outstanding performance on all datasets, with its average precision and recall rates reaching

Table 4: Overall Performance Comparison of Time Series Forecasting

| Models | | ETTm1 MSE | ETTm1 MAE | ETTm2 MSE | ETTm2 MAE | ETTh1 MSE | ETTh1 MAE | ETTh2 MSE | ETTh2 MAE | ECL MSE | ECL MAE | Weather MSE | Weather MAE | Traffic MSE | Traffic MAE | Solar MSE | Solar MAE |
|---|---|---|---|---|---|---|---|---|---|---|---|---|---|---|---|---|---|
| ARIMA | 96 | 0.893 | 0.694 | 2.071 | 1.103 | 1.074 | 0.803 | 2.552 | 1.308 | 0.405 | 0.467 | 0.399 | 0.436 | 0.873 | 0.483 | 1.260 | 1.352 |
| | 192 | 1.143 | 0.806 | 2.279 | 1.142 | 1.247 | 0.862 | 3.342 | 1.414 | 0.472 | 0.503 | 0.446 | 0.465 | 0.877 | 0.483 | 1.279 | 1.366 |
| | 336 | 1.297 | 0.862 | 2.598 | 1.268 | 1.289 | 0.871 | 3.321 | 1.418 | 0.469 | 0.503 | 0.485 | 0.484 | 0.883 | 0.485 | 1.300 | 1.381 |
| | 720 | 1.354 | 0.888 | 2.750 | 1.317 | 1.301 | 0.868 | 3.287 | 1.387 | 1.010 | 0.844 | 0.565 | 0.550 | 1.530 | 0.835 | 1.331 | 1.401 |
| | Avg | 1.172 | 0.813 | 2.425 | 1.208 | 1.228 | 0.851 | 3.126 | 1.382 | 0.589 | 0.579 | 0.474 | 0.484 | 1.041 | 0.572 | 1.293 | 1.375 |
| N-HiTS | 96 | 0.385 | 0.421 | 0.211 | 0.302 | 0.440 | 0.452 | 0.355 | 0.410 | 0.171 | 0.274 | 0.197 | 0.247 | 0.406 | 0.315 | 0.255 | 0.263 |
| | 192 | 0.417 | 0.458 | 0.274 | 0.349 | 0.470 | 0.481 | 0.423 | 0.456 | 0.192 | 0.296 | 0.248 | 0.296 | 0.424 | 0.330 | 0.270 | 0.284 |
| | 336 | 0.476 | 0.461 | 0.329 | 0.388 | 0.499 | 0.544 | 0.470 | 0.502 | 0.214 | 0.319 | 0.300 | 0.337 | 0.452 | 0.346 | 0.308 | 0.339 |
| | 720 | 0.531 | 0.503 | 0.405 | 0.442 | 0.564 | 0.577 | 0.494 | 0.511 | 0.263 | 0.361 | 0.371 | 0.387 | 0.544 | 0.387 | 0.306 | 0.343 |
| | Avg | 0.452 | 0.461 | 0.305 | 0.370 | 0.493 | 0.514 | 0.436 | 0.470 | 0.210 | 0.313 | 0.279 | 0.317 | 0.457 | 0.344 | 0.285 | 0.307 |
| TimesNet | 96 | 0.482 | 0.493 | 0.307 | 0.387 | 0.504 | 0.522 | 0.431 | 0.494 | 0.214 | 0.318 | 0.199 | 0.258 | 0.623 | 0.345 | 0.210 | 0.302 |
| | 192 | 0.504 | 0.507 | 0.369 | 0.429 | 0.556 | 0.549 | 0.522 | 0.534 | 0.222 | 0.325 | 0.252 | 0.299 | 0.626 | 0.347 | 0.229 | 0.316 |
| | 336 | 0.523 | 0.519 | 0.441 | 0.471 | 0.611 | 0.589 | 0.572 | 0.572 | 0.230 | 0.333 | 0.320 | 0.340 | 0.630 | 0.349 | 0.250 | 0.331 |
| | 720 | 0.590 | 0.564 | 0.528 | 0.523 | 0.641 | 0.620 | 0.582 | 0.588 | 0.258 | 0.355 | 0.406 | 0.394 | 0.649 | 0.365 | 0.281 | 0.351 |
| | Avg | 0.525 | 0.521 | 0.411 | 0.453 | 0.578 | 0.570 | 0.527 | 0.547 | 0.231 | 0.333 | 0.294 | 0.323 | 0.632 | 0.352 | 0.243 | 0.325 |
| PatchTST | 96 | 0.404 | 0.431 | 0.304 | 0.367 | 0.442 | 0.466 | 0.339 | 0.331 | 0.162 | 0.262 | 0.181 | 0.232 | 0.389 | 0.285 | 0.198 | 0.267 |
| | 192 | 0.435 | 0.452 | 0.371 | 0.412 | 0.468 | 0.490 | 0.342 | 0.336 | 0.181 | 0.280 | 0.224 | 0.275 | 0.407 | 0.295 | 0.219 | 0.287 |
| | 336 | 0.453 | 0.465 | 0.359 | 0.414 | 0.493 | 0.512 | 0.417 | 0.449 | 0.201 | 0.302 | 0.274 | 0.315 | 0.423 | 0.306 | 0.242 | 0.307 |
| | 720 | 0.464 | 0.490 | 0.409 | 0.452 | 0.517 | 0.540 | 0.474 | 0.505 | 0.252 | 0.348 | 0.347 | 0.368 | 0.466 | 0.335 | 0.270 | 0.335 |
| | Avg | 0.439 | 0.460 | 0.361 | 0.411 | 0.480 | 0.502 | 0.393 | 0.405 | 0.199 | 0.298 | 0.257 | 0.298 | 0.421 | 0.305 | 0.232 | 0.299 |
| DLinear | 96 | 0.405 | 0.429 | 0.319 | 0.383 | 0.413 | 0.443 | 0.286 | 0.360 | 0.168 | 0.268 | 0.199 | 0.259 | 0.429 | 0.315 | 0.223 | 0.288 |
| | 192 | 0.435 | 0.446 | 0.413 | 0.448 | 0.453 | 0.465 | 0.343 | 0.403 | 0.182 | 0.281 | 0.241 | 0.298 | 0.439 | 0.320 | 0.244 | 0.304 |
| | 336 | 0.469 | 0.473 | 0.478 | 0.495 | 0.489 | 0.486 | 0.394 | 0.442 | 0.197 | 0.298 | 0.288 | 0.336 | 0.452 | 0.327 | 0.263 | 0.321 |
| | 720 | 0.502 | 0.520 | 0.635 | 0.581 | 0.536 | 0.521 | 0.482 | 0.521 | 0.233 | 0.332 | 0.350 | 0.392 | 0.491 | 0.349 | 0.276 | 0.337 |
| | Avg | 0.453 | 0.467 | 0.461 | 0.477 | 0.473 | 0.479 | 0.376 | 0.432 | 0.195 | 0.295 | 0.270 | 0.321 | 0.453 | 0.328 | 0.252 | 0.313 |
| UniTime | 96 | 0.390 | 0.439 | 0.320 | 0.386 | 0.594 | 0.598 | 0.433 | 0.465 | 0.159 | 0.296 | 0.183 | 0.253 | 0.373 | 0.321 | 0.201 | 0.304 |
| | 192 | 0.418 | 0.607 | 0.388 | 0.430 | 0.571 | 0.559 | 0.511 | 0.514 | 0.177 | 0.408 | 0.231 | 0.481 | 0.392 | 0.383 | 0.220 | 0.461 |
| | 336 | 0.431 | 0.710 | 0.516 | 0.531 | 0.665 | 0.637 | 0.612 | 0.607 | 0.192 | 0.478 | 0.286 | 0.600 | 0.409 | 0.419 | 0.233 | 0.566 |
| | 720 | 0.436 | 0.749 | 0.657 | 0.630 | 0.706 | 0.697 | 0.662 | 0.664 | 0.229 | 0.532 | 0.361 | 0.623 | 0.443 | 0.465 | 0.252 | 0.638 |
| | Avg | 0.419 | 0.626 | 0.470 | 0.494 | 0.634 | 0.623 | 0.555 | 0.563 | 0.189 | 0.429 | 0.265 | 0.489 | 0.404 | 0.397 | 0.227 | 0.492 |
| UniTS | 96 | 0.407 | 0.434 | 0.336 | 0.400 | 0.416 | 0.390 | 0.284 | 0.307 | 0.167 | 0.260 | 0.184 | 0.235 | 0.425 | 0.313 | 0.226 | 0.291 |
| | 192 | 0.442 | 0.456 | 0.430 | 0.465 | 0.452 | 0.412 | 0.341 | 0.350 | 0.188 | 0.288 | 0.226 | 0.279 | 0.455 | 0.332 | 0.249 | 0.314 |
| | 336 | 0.470 | 0.474 | 0.495 | 0.512 | 0.486 | 0.433 | 0.398 | 0.389 | 0.211 | 0.315 | 0.274 | 0.312 | 0.493 | 0.358 | 0.275 | 0.341 |
| | 720 | 0.518 | 0.513 | 0.652 | 0.598 | 0.542 | 0.468 | 0.514 | 0.468 | 0.288 | 0.385 | 0.348 | 0.365 | 0.590 | 0.422 | 0.315 | 0.384 |
| | Avg | 0.459 | 0.469 | 0.478 | 0.494 | 0.474 | 0.426 | 0.384 | 0.379 | 0.214 | 0.312 | 0.258 | 0.298 | 0.491 | 0.356 | 0.266 | 0.333 |
| iTransformer | 96 | 0.364 | 0.398 | 0.229 | 0.297 | 0.416 | 0.427 | 0.336 | 0.388 | 0.165 | 0.261 | 0.194 | 0.231 | 0.428 | 0.287 | 0.241 | 0.247 |
| | 192 | 0.394 | 0.434 | 0.259 | 0.353 | 0.444 | 0.455 | 0.400 | 0.432 | 0.190 | 0.260 | 0.268 | 0.273 | 0.444 | 0.281 | 0.254 | 0.267 |
| | 336 | 0.450 | 0.436 | 0.327 | 0.351 | 0.470 | 0.517 | 0.444 | 0.476 | 0.188 | 0.295 | 0.315 | 0.342 | 0.474 | 0.300 | 0.291 | 0.321 |
| | 720 | 0.502 | 0.476 | 0.449 | 0.430 | 0.534 | 0.548 | 0.468 | 0.484 | 0.245 | 0.329 | 0.378 | 0.379 | 0.493 | 0.351 | 0.289 | 0.325 |
| | Avg | 0.428 | 0.436 | 0.316 | 0.358 | 0.466 | 0.487 | 0.412 | 0.445 | 0.197 | 0.286 | 0.289 | 0.306 | 0.460 | 0.305 | 0.269 | 0.290 |
| TimeVLM | 96 | 0.316 | 0.368 | 0.186 | 0.286 | 0.399 | 0.412 | 0.287 | 0.350 | 0.162 | 0.272 | 0.183 | 0.239 | 0.419 | 0.314 | 0.189 | 0.251 |
| | 192 | 0.355 | 0.400 | 0.244 | 0.326 | 0.428 | 0.440 | 0.350 | 0.402 | 0.176 | 0.282 | 0.228 | 0.275 | 0.445 | 0.333 | 0.231 | 0.268 |
| | 336 | 0.382 | 0.422 | 0.306 | 0.369 | 0.448 | 0.457 | 0.374 | 0.416 | 0.192 | 0.297 | 0.276 | 0.311 | 0.459 | 0.338 | 0.276 | 0.315 |
| | 720 | 0.431 | 0.450 | 0.419 | 0.441 | 0.486 | 0.494 | 0.445 | 0.471 | 0.231 | 0.326 | 0.342 | 0.357 | 0.530 | 0.380 | 0.290 | 0.352 |
| | Avg | 0.371 | 0.410 | 0.289 | 0.355 | 0.440 | 0.451 | 0.364 | 0.410 | 0.190 | 0.294 | 0.257 | 0.296 | 0.463 | 0.341 | 0.247 | 0.297 |
| AutoTimes | 96 | 0.416 | 0.430 | 0.303 | 0.403 | 0.577 | 0.615 | 0.416 | 0.482 | 0.201 | 0.255 | 0.210 | 0.233 | 0.468 | 0.278 | 0.253 | 0.251 |
| | 192 | 0.429 | 0.449 | 0.371 | 0.447 | 0.554 | 0.576 | 0.494 | 0.531 | 0.323 | 0.271 | 0.480 | 0.280 | 0.568 | 0.387 | 0.403 | 0.266 |
| | 336 | 0.444 | 0.459 | 0.499 | 0.548 | 0.648 | 0.654 | 0.595 | 0.624 | 0.409 | 0.288 | 0.624 | 0.323 | 0.651 | 0.396 | 0.475 | 0.278 |
| | 720 | 0.461 | 0.470 | 0.640 | 0.647 | 0.689 | 0.714 | 0.645 | 0.681 | 0.485 | 0.318 | 0.648 | 0.375 | 0.767 | 0.414 | 0.556 | 0.292 |
| | Avg | 0.438 | 0.452 | 0.453 | 0.511 | 0.617 | 0.640 | 0.538 | 0.580 | 0.355 | 0.283 | 0.491 | 0.303 | 0.614 | 0.369 | 0.422 | 0.272 |
| TimeLLM | 96 | 0.410 | 0.442 | 0.343 | 0.418 | 0.617 | 0.630 | 0.456 | 0.497 | 0.167 | 0.274 | 0.179 | 0.230 | 0.406 | 0.310 | 0.254 | 0.319 |
| | 192 | 0.438 | 0.461 | 0.411 | 0.462 | 0.594 | 0.591 | 0.534 | 0.546 | 0.188 | 0.296 | 0.223 | 0.273 | 0.427 | 0.324 | 0.278 | 0.345 |
| | 336 | 0.455 | 0.473 | 0.539 | 0.563 | 0.688 | 0.669 | 0.635 | 0.639 | 0.213 | 0.322 | 0.273 | 0.314 | 0.450 | 0.341 | 0.299 | 0.368 |
| | 720 | 0.464 | 0.493 | 0.680 | 0.662 | 0.729 | 0.729 | 0.685 | 0.696 | 0.277 | 0.378 | 0.345 | 0.362 | 0.478 | 0.356 | 0.340 | 0.426 |
| | Avg | 0.442 | 0.467 | 0.493 | 0.526 | 0.657 | 0.655 | 0.578 | 0.595 | 0.211 | 0.318 | 0.255 | 0.295 | 0.440 | 0.333 | 0.293 | 0.365 |
| Timer | 96 | 0.322 | 0.372 | 0.208 | 0.290 | 0.371 | 0.411 | 0.302 | 0.362 | 0.164 | 0.261 | 0.188 | 0.236 | 0.329 | 0.255 | 0.183 | 0.234 |
| | 192 | 0.367 | 0.406 | 0.265 | 0.334 | 0.411 | 0.438 | 0.359 | 0.404 | 0.184 | 0.281 | 0.235 | 0.283 | 0.347 | 0.260 | 0.333 | 0.391 |
| | 336 | 0.400 | 0.431 | 0.315 | 0.372 | 0.435 | 0.454 | 0.387 | 0.429 | 0.204 | 0.301 | 0.284 | 0.320 | 0.363 | 0.267 | 0.405 | 0.496 |
| | 720 | 0.448 | 0.463 | 0.390 | 0.420 | 0.463 | 0.488 | 0.431 | 0.472 | 0.245 | 0.337 | 0.345 | 0.366 | 0.406 | 0.289 | 0.486 | 0.568 |
| | Avg | 0.384 | 0.418 | 0.295 | 0.354 | 0.420 | 0.448 | 0.370 | 0.417 | 0.199 | 0.295 | 0.263 | 0.301 | 0.361 | 0.268 | 0.352 | 0.422 |
| TimerXL | 96 | 0.310 | 0.364 | 0.200 | 0.286 | 0.378 | 0.415 | 0.301 | 0.363 | 0.162 | 0.259 | 0.187 | 0.235 | 0.369 | 0.253 | 0.184 | 0.249 |
| | 192 | 0.351 | 0.396 | 0.259 | 0.330 | 0.423 | 0.448 | 0.357 | 0.406 | 0.182 | 0.280 | 0.235 | 0.281 | 0.379 | 0.272 | 0.208 | 0.275 |
| | 336 | 0.380 | 0.419 | 0.311 | 0.367 | 0.452 | 0.470 | 0.384 | 0.432 | 0.203 | 0.301 | 0.283 | 0.319 | 0.392 | 0.298 | 0.229 | 0.298 |
| | 720 | 0.424 | 0.448 | 0.381 | 0.417 | 0.511 | 0.523 | 0.411 | 0.465 | 0.248 | 0.341 | 0.350 | 0.366 | 0.431 | 0.362 | 0.270 | 0.356 |
| | Avg | 0.366 | 0.407 | 0.288 | 0.350 | 0.441 | 0.464 | 0.363 | 0.417 | 0.199 | 0.295 | 0.264 | 0.300 | 0.393 | 0.296 | 0.223 | 0.295 |
| TimeCDS | 96 | 0.299 | 0.347 | 0.178 | 0.265 | 0.342 | 0.383 | 0.272 | 0.330 | 0.151 | 0.235 | 0.163 | 0.228 | 0.365 | 0.261 | 0.169 | 0.184 |
| | 192 | 0.335 | 0.379 | 0.234 | 0.297 | 0.387 | 0.409 | 0.334 | 0.373 | 0.167 | 0.268 | 0.211 | 0.256 | 0.395 | 0.323 | 0.175 | 0.196 |
| | 336 | 0.375 | 0.412 | 0.275 | 0.339 | 0.416 | 0.426 | 0.361 | 0.400 | 0.186 | 0.282 | 0.264 | 0.304 | 0.433 | 0.359 | 0.194 | 0.208 |
| | 720 | 0.415 | 0.434 | 0.343 | 0.378 | 0.433 | 0.459 | 0.394 | 0.457 | 0.224 | 0.317 | 0.329 | 0.347 | 0.530 | 0.405 | 0.228 | 0.222 |
| | Avg | 0.356 | 0.393 | 0.258 | 0.320 | 0.395 | 0.419 | 0.340 | 0.390 | 0.182 | 0.262 | 0.242 | 0.284 | 0.431 | 0.337 | 0.192 | 0.203 |

81.02% and 83.64%, respectively. This result indicates that TimeCDS outperforms other models such as Informer and TimesNet in terms of the accuracy and comprehensiveness of anomaly detection. Their average precision and recall rates are 68.45%, 67.31% and 75.82%, 71.31%, respectively. In addition, the impact of different datasets on model performance is also worthy of attention. For instance, the overall performance of the model on the SMAP dataset is superior to that on the SWaT dataset, which may be related to the specific characteristics of the dataset, such as data size and abnormal distribution. The consistent high-performance performance of TimeCDS on different datasets further demonstrates its excellent generalization ability. Overall, the performance of TimeCDS in time series anomaly detection tasks not only validates its effectiveness but also provides valuable references for research and practice in related fields.

Table 7: Overall performance of time series classification without pre-training

| Metric | Informer | FEDformer | Full Attn | LIGHTTS | DLiner | LSTM | LSTNET | TimesNet | UniTS | Timer | TimeCDS |
|---|---|---|---|---|---|---|---|---|---|---|---|
| Accuracy | 0.678 | 0.729 | 0.721 | 0.702 | 0.736 | 0.521 | 0.663 | 0.740 | 0.726 | 0.752 | 0.762 |
| F1 | 0.662 | 0.634 | 0.688 | 0.674 | 0.709 | 0.569 | 0.658 | 0.717 | 0.693 | 0.718 | 0.723 |
| Recall | 0.632 | 0.696 | 0.684 | 0.711 | 0.701 | 0.598 | 0.685 | 0.698 | 0.696 | 0.703 | 0.713 |
| AUC | 0.659 | 0.682 | 0.677 | 0.666 | 0.677 | 0.519 | 0.649 | 0.686 | 0.679 | 0.696 | 0.687 |

## C.3 TIME SERIES CLASSIFICATION

Table 7 details the performance of each model in the time series classification task, supplementing the two key indicators of Accuracy and Recall. These indicators are crucial for a comprehensive assessment of the model's performance in classification tasks. The TimeCDS model demonstrates outstanding performance in all four major evaluation metrics. Specifically, TimeCDS achieved an accuracy rate of 0.762 and a recall rate of 0.713, both significantly outperformed other models. This indicates that TimeCDS has significant advantages in correctly classifying the proportion of samples and identifying all positive category samples. In addition, TimeCDS also performed the best in F1 score (0.723) and AUC value (0.687), further demonstrating its excellent balance between precision and recall, and its strong ability to distinguish between positive and negative class samples. The Timer and TimesNet models also performed well, achieving results of 0.752 and 0.718 in accuracy and 0.740 and 0.698 in recall rates, respectively. These results indicate that although slightly inferior to TimeCDS, these models still have high performance in time series classification tasks. Overall, the TimeCDS model performs comprehensively and evenly in time series classification tasks. It significantly outperforms other models in terms of accuracy, recall rate, F1 score, and AUC value. This result not only verifies the validity of the TimeCDS model, but also provides valuable references for research and practice in related fields. The outstanding performance of TimeCDS indicates that it can effectively capture the features of data and make accurate classification decisions when dealing with time series classification problems, which holds significant value in practical applications.

## C.4 COMPARSION WITH SOTA

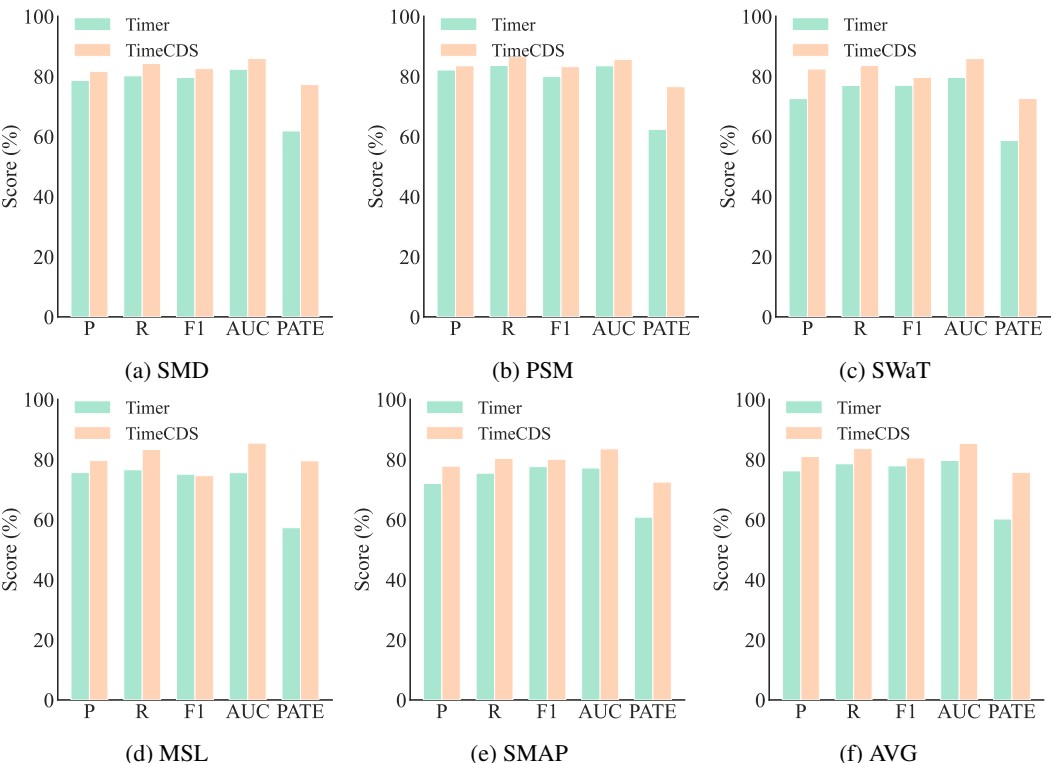

Figure 8: Anomaly detection comparison: TimeCDS vs Timer

Figure 9 visually shows that TimeCDS outperforms the Timer model in all indicators, which powerfully demonstrates the superiority of TimeCDS.

## C.5 EFFICIENCY

In this study, we conducted an in-depth analysis of the computational efficiency of the model to optimize resource utilization and enhance performance. By introducing the FlashAttention strategy, we significantly reduced the computational complexity of the model. Specifically, we conducted a detailed assessment of each module of the model in terms of FLOPs, parameter count, and memory usage. In the complexity expression, N represents the total scale of the time series data, T is the

Table 8: Analysis of the computational complexity of each module of the model

| Module | FLOPs | Parameters | Memory Footprint |
|---|---|---|---|
| Channel Dependency Search | $O(NlogN+K^2)$ | $O(K^2)$ | $O(NT+NlogN+K^2)$ |
| Time Series Encoding Branch | $O(BK(T+MLd))$ | $O(K^2d+Kd^2)$ | $O(BK(T+ML+Md))$ |
| Time Image Encoding Branch | $O(BK(T+LK+LD))$ | $O(FKLT+K^2)$ | $O(FKLT+K^2+KD)$ |
| Cross-Modal Alignment | $O(BL(d^2+K))$ | $O(Kd)$ | $O(BL(d+L))$ |

length of a single time series, K is the number of representative features selected, B is the batch size, M is the number of blocks the time series is divided into, L is the length of each block, d is the feature dimension, and F is the size of the convolution kernel. D is related to the parameters of the projection layer. These symbols jointly describe the computational overhead of the model at different operation stages. As shown in Table 9, compared with the baseline model, our model

Table 9: Model efficiency performance comparison

| Model | Training Speed (s/iter) | Inference Speed (s/iter) | Params (M) | Performance (MSE) |
|---|---|---|---|---|
| TimeCDS | 0.2426 | 0.0454 | 16.73 | 0.151 |
| PatchTST | 0.0625 | 0.0079 | 16.21 | 0.162 |
| Timer-XL | 0.2157 | 0.0447 | 15.86 | 0.164 |
| Timer | 0.0458 | 0.0079 | 15.86 | 0.162 |

achieves higher operational efficiency while maintaining a lower computational cost. For instance, the TimeCDS model achieved a training speed of 0.2426 seconds per iteration and an inference speed of 0.0454 seconds per iteration, while maintaining a parameter count of 16.73M and a mean square error (MSE) performance metric of 0.151. These results demonstrate the effectiveness and efficiency of our model in processing high-dimensional time series data.

## C.6 PARAMETER SENSITIVITY

To further demonstrate the validity of the model, we chose to evaluate the hyperparameter sensitivity of TimeCDS on the widely recognized ERA 5-MS benchmark. The main ones are the patch size P and the Lookback Length L during the inference process. Our research results show that the optimal patch size is usually close to the predicted length because it avoids multi-step error accumulation. Meanwhile, the research on Lookback Length L found that the optimal lookback length is not necessarily the same as the length used during training, indicating that the appropriate selection of information length is effective, and the reasoning stage can be compatible with different lengths and dimensions of cross-domain time series data.

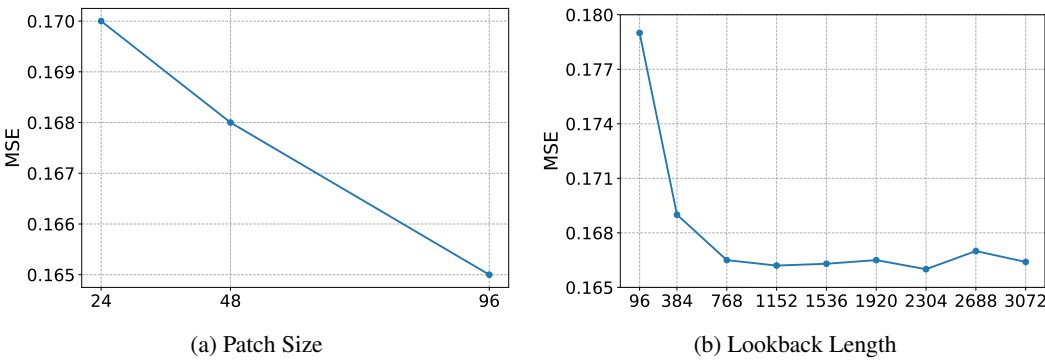

(a) Patch Size              (b) Lookback Length

Figure 9: Sensitive analysis of Patch Size and Lookback Length

