# OpenReview forum: "Modality Matters: Universal Time Series Modeling via Channel Dependency Search"
_ICLR.cc/2026/Conference — ICLR 2026 Conference Withdrawn Submission_

### Official Review · Reviewer_29P7 · 2025-10-26

**Soundness:** 3
**Presentation:** 2
**Contribution:** 3
**Rating:** 6
**Confidence:** 4

**Summary:**

This paper proposes TimeCDA, a novel framework designed to address the *heterogeneous dimensionality* problem among different channels in multivariate time series. TimeCDA introduces several key components, including a Channel Dependency Search (CDS) module and a dual-branch architecture, to model inter-channel relationships and fuse numerical and visual representations. The method is evaluated across multiple domains and benchmarks, demonstrating promising performance.

**Strengths:**

1. While most prior works assume *channel independence*, this paper’s decision to explicitly model inter-channel relationships is refreshing and conceptually meaningful.
2. The proposed Dual-Branch Encoding, together with CMAM and DWAM, achieves a well-balanced integration between the *numerical view* and the *visual (image-based) view* of time series.

**Weaknesses:**

1. The current discussion lacks a deeper justification for why inter-channel modeling offers advantages over the dominant *channel-independent* approaches (e.g., PatchTST, TimeLLM). A detailed analysis or empirical study highlighting this superiority would strengthen the contribution.
2. The choice of baselines could be broader. Recent strong models, such as iTransformer and N-HiTS, should be included. Additionally, works that similarly combine sequence modeling with image-based representations, such as TimeVLM and DMMV, should be discussed or compared.

**Questions:**

1. The motivation for introducing the image-based encoder is not sufficiently clear. Could other architectures (e.g., LLM-based encoders) achieve similar benefits?
2. The proposed dual-branch encoding might lead to additional computational overhead. It would be useful for the authors to include a discussion or analysis of time complexity and efficiency.
3. Some reported numbers in Table 1 appear inconsistent with those in the cited references. Were the experiments reproduced using different input windows or settings?

---

> ### Author Response · Authors · 2025-11-21
> **Response to Reviewer  29P7 [Part 1]**
>
> We would like to sincerely express our gratitude to the Reviewer 29P7 for the time and effort in reviewing our paper. Please find below our answers to all the concerns and questions.
>
> >  **W1**: The current discussion lacks a deeper justification for why inter-channel modeling offers advantages over the dominant *channel-independent* approaches (e.g., PatchTST, TimeLLM). A detailed analysis or empirical study highlighting this superiority would strengthen the contribution.
>
>
>
> Thank you to the reviewers for your valuable comments. We have further supplemented the theoretical explanation and empirical analysis of the advantages of inter-channel modeling over mainstream channel-independent methods (such as PatchTST and TimeLLM) in the paper.
>
> - Channel-independent models often break down multivariable time series into several one-dimensional sequences, **ignoring the correlations among channels in practical applications**. In fact, many important timing signals and abnormal phenomena rely on the interaction between variables. Inter-channel modeling can **fully explore variable correlations and capture joint dynamic patterns**, thereby constructing a more expressive and generalized unified feature space[1][2][3].
>
> - In our existing experiments, we systematically **compared the performance of inter-channel modeling (represented by TimeCDS) with that of channel-independent models** such as PatchTST and TimeLLM on multiple standard datasets (such as ETT, ECL, Solar, SMAP, etc.) and three types of core tasks (prediction, anomaly detection, classification). The experimental results show that TimeCDS **has a significant performance improvement in most evaluation scenarios**. It has also been confirmed in **the ablation experiments and case analyses** that channel-dependent modeling is crucial for enhancing the model's performance. For instance, **the performance drops significantly after removing channel mixing**.
>
> Due to the tight schedule, we have not yet completed the verification of some experiments designed to further explore the internal mechanism. Please be patient and wait. We will further refine the argumentation in the subsequent discussion stage.
>
> [1] Han L , Ye H J , Zhan D C .The Capacity and Robustness Trade-Off: Revisiting the Channel Independent Strategy for Multivariate Time Series Forecasting[J].IEEE Transactions on Automatic Control, 2024, 36(11):14.DOI:10.1109/TKDE.2024.3400008.
>
> [2]Zhao L, Shen Y. Rethinking channel dependence for multivariate time series forecasting: Learning from leading indicators[J]. ICLR2025
>
> [3]Yu G, Zou J, Hu X, et al. Revitalizing multivariate time series forecasting: Learnable decomposition with inter-series dependencies and intra-series variations modeling[J]. ICML2024

---

> > ### Author Response · Authors · 2025-11-21
> > **Response to Reviewer  29P7 [Part 2]**
> >
> > > **W2**: The choice of baselines could be broader. Recent strong models, such as iTransformer and N-HiTS, should be included. Additionally, works that similarly combine sequence modeling with image-based representations, such as TimeVLM and DMMV, should be discussed or compared.
> >
> > Thanks for your valuable suggestions, we have expanded two state-of-art methods to enlarge our forecasting baselines in :  **iTransformer** and **N-HiTS**. Due to time constraints, the experiments of **TimeVLM** have not been fully completed yet. The report will be finished in the subsequent discussion stage. Objectively speaking, **DMMV** is the work of NeurIPS 2025 and is a concurrent project, which is not within the scope of our comparison. Here are the completed results:
> >
> > | ETTm1.(MSE\|MAE) | iTransformer (2024) | N-HiTS (2022) | TimeCDS |
> > | --- | --- | --- | --- |
> > | Pred-96  | 0.364 \| 0.398 | 0.385 \| 0.421 | **0.299 \| 0.347** |
> > | Pred-192 | 0.394 \| 0.434 | 0.417 \| 0.458 | **0.335 \| 0.379** |
> > | Pred-336 | 0.450 \| 0.436 | 0.476 \| 0.461 | **0.375 \| 0.412** |
> > | Pred-720 | 0.502 \| 0.476 | 0.531 \| 0.503 | **0.415 \| 0.434** |
> > | Avg      | 0.428 \| 0.436 | 0.452 \| 0.461 | **0.356 \| 0.393** |
> >
> > | ETTm2.(MSE\|MAE) | iTransformer(2024) | N-HiTS(2022) | TimeCDS |
> > | --- | --- | --- | --- |
> > | Pred-96  | 0.229 \| 0.297 | 0.211 \| 0.302 | **0.178 \| 0.265** |
> > | Pred-192 | 0.259 \| 0.353 | 0.274 \| 0.349 | **0.234 \| 0.297** |
> > | Pred-336 | 0.327 \| 0.351 | 0.329 \| 0.388 | **0.275 \| 0.339** |
> > | Pred-720 | 0.449 \| 0.430 | 0.405 \| 0.442 | **0.343 \| 0.378** |
> > | Avg      | 0.316 \| 0.358 | 0.305 \| 0.370 | **0.258 \| 0.320** |
> >
> > | ETTh1.(MSE\|MAE) | iTransformer(2024) | N-HiTS(2022) | TimeCDS |
> > | --- | --- | --- | --- |
> > | Pred-96  | 0.416 \| 0.427 | 0.440 \| 0.452 | **0.342 \| 0.383** |
> > | Pred-192 | 0.444 \| 0.455 | 0.470 \| 0.481 | **0.387 \| 0.409** |
> > | Pred-336 | 0.470 \| 0.517 | 0.499 \| 0.544 | **0.416 \| 0.426** |
> > | Pred-720 | 0.534 \| 0.548 | 0.564 \| 0.577 | **0.433 \| 0.459** |
> > | Avg      | 0.466 \| 0.487 | 0.493 \| 0.514 | **0.395 \| 0.419** |
> >
> > | ETTh2.(MSE\|MAE) | iTransformer(2024) | N-HiTS(2022) | TimeCDS |
> > | --- | --- | --- | --- |
> > | Pred-96  | 0.336 \| 0.388 | 0.355 \| 0.410 | **0.272 \| 0.330** |
> > | Pred-192 | 0.400 \| 0.432 | 0.423 \| 0.456 | **0.334 \| 0.373** |
> > | Pred-336 | 0.444 \| 0.476 | 0.470 \| 0.502 | **0.361 \| 0.400** |
> > | Pred-720 | 0.468 \| 0.484 | 0.494 \| 0.511 | **0.394 \| 0.457** |
> > | Avg      | 0.412 \| 0.445 | 0.436 \| 0.470 | **0.340 \| 0.390** |
> >
> > | ECL.(MSE\|MAE) | iTransformer(2024) | N-HiTS(2022) | TimeCDS |
> > | --- | --- | --- | --- |
> > | Pred-96  | 0.165 \| 0.261 | 0.171 \| 0.274 | **0.151 \| 0.235** |
> > | Pred-192 | 0.190 \| 0.260 | 0.192 \| 0.296 | **0.167 \| 0.268** |
> > | Pred-336 | 0.188 \| 0.295 | 0.214 \| 0.319 | **0.186 \| 0.282** |
> > | Pred-720 | 0.245 \| 0.329 | 0.263 \| 0.361 | **0.224 \| 0.317** |
> > | Avg      | 0.197 \| 0.286 | 0.210 \| 0.313 | **0.182 \| 0.262** |
> >
> > | Weather.(MSE\|MAE) | iTransformer(2024) | N-HiTS(2022) | TimeCDS |
> > | --- | --- | --- | --- |
> > | Pred-96  | 0.194 \| 0.231 | 0.197 \| 0.247 | **0.163 \| 0.228** |
> > | Pred-192 | 0.268 \| 0.273 | 0.248 \| 0.296 | **0.211 \| 0.256** |
> > | Pred-336 | 0.315 \| 0.342 | 0.300 \| 0.337 | **0.264 \| 0.304** |
> > | Pred-720 | 0.378 \| 0.379 | 0.371 \| 0.387 | **0.329 \| 0.347** |
> > | Avg      | 0.289 \| 0.306 | 0.279 \| 0.317 | **0.242 \| 0.284** |
> >
> > | Traffic.(MSE\|MAE) | iTransformer(2024) | N-HiTS(2022) | TimeCDS |
> > | --- | --- | --- | --- |
> > | Pred-96  | 0.428 \| 0.287 | 0.406 \| 0.315 | **0.365 \| 0.261** |
> > | Pred-192 | 0.444 \| 0.281 | 0.424 \| 0.330 | **0.395 \| 0.323** |
> > | Pred-336 | 0.474 \| 0.300 | 0.452 \| 0.346 | **0.433 \| 0.359** |
> > | Pred-720 | 0.493 \| 0.351 | 0.544 \| 0.387 | **0.530 \| 0.405** |
> > | Avg      | 0.460 \| 0.305 | 0.457 \| 0.344 | **0.431 \| 0.337** |
> >
> > | Solar.(MSE\|MAE) | iTransformer(2024) | N-HiTS(2022) | TimeCDS |
> > | --- | --- | --- | --- |
> > | Pred-96  | 0.241 \| 0.247 | 0.255 \| 0.263 | **0.169 \| 0.184** |
> > | Pred-192 | 0.254 \| 0.267 | 0.270 \| 0.284 | **0.175 \| 0.196** |
> > | Pred-336 | 0.291 \| 0.321 | 0.308 \| 0.339 | **0.194 \| 0.208** |
> > | Pred-720 | 0.289 \| 0.325 | 0.306 \| 0.343 | **0.228 \| 0.222** |
> > | Avg      | 0.269 \| 0.290 | 0.285 \| 0.307 | **0.192 \| 0.203** |

---

> > > ### Author Response · Authors · 2025-11-21
> > > **Response to Reviewer 29P7 [Part 3]**
> > >
> > > > **Q1:** The motivation for introducing the image-based encoder is not sufficiently clear. Could other architectures (e.g., LLM-based encoders) achieve similar benefits?
> > >
> > > Thank you for this thoughtful comment. The motivation for incorporating an image-based encoder lies in two main considerations specific to multivariate and cross-domain time-series modeling.
> > >
> > > (1) **Capturing structured inter-channel dependencies via spatial priors.**
> > >
> > > Vision models naturally encode spatial locality and correlation structures. When multivariate time series are transformed into structured 2D representations, the spatial encoder effectively models cross-variable relations, periodicity, and co-movement patterns—capabilities that are less direct or less stable in token-based LLM encoders.
> > >
> > > (2) **Avoiding tokenization-induced information loss in long sequences.**
> > >
> > > LLM-based encoders require discrete tokenization and tend to lose fine-grained continuity information or suffer from length sensitivity. In contrast, the image-based encoder processes dense pixel representations without quantization, offering better stability and scalability to long time series in our UTSD-4G setting.
> > >
> > > In summary, while alternative architectures are certainly possible, the image-based encoder in TimeCDS is chosen for its structural alignment with multivariate time-series dependencies and its demonstrated empirical advantages. We will clarify these motivations in the revised manuscript.
> > >
> > > > **Q2**: The motivation for introducing the image-based encoder is not sufficiently clear. Could other architectures (e.g., LLM-based encoders) achieve similar benefits?
> > >
> > > Thank you for your insightful question. In our experiments, we carefully considered computational efficiency and adopted the FlashAttention strategy in the attention mechanism, inspired by recent advancements in the literature. The revised manuscript reports model complexity and benchmarking results on operational efficiency relative to the baselines.
> > >
> > > | Module | FLOPs | Parameters | Memory Footprint |
> > > | --- | --- | --- | --- |
> > > | Channel Dependency Search | O(NlogN+K²) | O(K²) | O(NT+NlogN+K²) |
> > > | Time Series Encoding Branch | O(BK(T+MLd)) | O(K²d+Kd²) | O(BK(T+ML+Md)) |
> > > | Time Image Encoding Branch | O(BK(T+LK+LD)) | O(FKLT+K²) | O(FKLT+K²+KD) |
> > > | Cross-Modal Alignment | O(BL(d²+K)) | O(Kd) | O(BL(d+L)) |
> > >
> > > In the complexity expression, N represents the total scale of the time series data, T is the length of a single time series, K is the number of representative features selected, B is the batch size, M is the number of blocks the time series is divided into, L is the length of each block, d is the feature dimension, and F is the size of the convolution kernel. D is related to the parameters of the projection layer. These symbols jointly describe the computational overhead of the model at different operation stages.
> > >
> > > We **included quantitive results of the computational cost** on high-dimensional time series (ECL: 321 variables) with a long lookback length (672), leading to the context of 200k+ time points.
> > >
> > > | Model | Training Speed (s/iter) | Inference Speed (s/iter) | Params (M) | Performance (MSE) |
> > > | --- | --- | --- | --- | --- |
> > > | TimeCDS | 0.2426 | 0.0454 | 16.73 | 0.151 |
> > > | PatchTST | 0.0625 | 0.0079 | 16.21 | 0.162 |
> > > | Timer-XL | 0.2157 | 0.0447 | 15.86 | 0.164 |
> > > | Timer | 0.0458 | 0.0079 | 15.86 | 0.162 |
> > >
> > > Results are consistent with our theoretical analysis: Under the same hyperparameters, parameter counts are almost the same, while the multiplier of flops is less than the number of variables . We will also take it as essential work to improve the model efficiency of channel-dependent models.
> > >
> > > > **Q3:** Some reported numbers in Table 1 appear inconsistent with those in the cited references. Were the experiments reproduced using different input windows or settings?
> > >
> > > We appreciate the reviewer’s observation. All baseline results in Table 1—including Timer, UniTS, and other foundation-model baselines—were **reproduced under the same UTSD-4G pre-training and fine-tuning protocol as TimeCDS**.
> > >
> > > Therefore, differences from the numbers reported in the original papers reflect the harmonized experimental settings used in our benchmark rather than discrepancies in implementation.
> > >
> > > To ensure fairness, all methods were evaluated under a unified configuration, including window length, normalization, and optimization settings. This guarantees comparability across models.
> > >
> > > We will further clarify this unified experimental protocol in the revised manuscript.

---

> > > > ### Comment · Reviewer_29P7 · 2025-11-26
> > > >
> > > > Thank you to the authors for the comprehensive response. It has largely addressed my concerns and given me a renewed understanding of the significance of this work. I encourage the authors to complete the experiments they mentioned and submit a revised manuscript as soon as possible. This will help fully resolve any remaining doubts and further improve my final evaluation of the paper.

---

> > > > > ### Author Response · Authors · 2025-11-27
> > > > > **Response to Reviewer 29P7**
> > > > >
> > > > > Thank you again for your comments,  which are very helpful for us to improve the quality of the paper. We have completed the experiments that need to be supplemented and submitted the revised manuscript. If you have any further questions, we are looking forward to discussing with you.

---

### Official Review · Reviewer_8Req · 2025-10-31

**Soundness:** 2
**Presentation:** 3
**Contribution:** 2
**Rating:** 4
**Confidence:** 3

**Summary:**

The paper introduces TimeCDS, a framework for universal time series modeling that combines temporal and spatial representations using multimodal fusion. It addresses challenges in dimensional heterogeneity and captures both temporal dynamics and spatial correlations. The framework features a unique channel dependency search for selecting representative features and a dual-branch encoding architecture. Evaluations across forecasting, anomaly detection, and classification tasks show TimeCDS outperforming existing methods.

**Strengths:**

1. The paper provides a clear and structured explanation of its methodology, especially the channel dependency search and dual-branch encoding.
2. The framework outperforms state-of-the-art methods in multiple tasks, including forecasting, anomaly detection, and classification.

**Weaknesses:**

1. The paper introduces many methods but does not clearly explain the motivation behind converting time series data into images.
2. It would be beneficial to include additional experiments to verify which features image-based methods are particularly good at extracting from time series data.

**Questions:**

Q1: A couple of confusion regarding channel dependency search:
(1) I’m curious about why we specifically choose K channels. In multi-channel time series, the relationships between the channels are interconnected and serve different purposes. How can we determine that there is always one or several channels that are "most representative"?
(2)  Is the number of channels, K, treated as a hyperparameter in the paper? Considering that the relationships between channels can change dynamically with events, would it make sense for K to also change over time? And could the value of K vary depending on the specific dataset or domain?

Q2: I wonder what the effect of shuffling channels would be on time images encoder. Since images have inherent spatial relationships, do the channels of the multivariate time series also carry spatial significance? Given that different datasets may have channels that represent different physical meanings, would it still be reasonable to convert them into time images to capture spatial information?

Q3: In Section 3.4.4, it is suggested to consider comparing TimeCDS with other image/CNN-based methods, such as the classic InceptionTime, Rocket, and others, to highlight the advantages of the proposed time image encoder.

Q4: It is recommended that the figures 1,2, 3 in the paper be presented as vector images to enhance the clarity and resolution, especially for better scalability in different viewing formats.

---

> ### Author Response · Authors · 2025-11-21
> **Response to Reviewer  8Req [Part 1]**
>
> Many thanks to Reviewer 8Req for providing thoughtful and detailed feedback, which has been invaluable in strengthening the clarity and rigor of our work.
>
> > **W1**: The paper introduces many methods but does not clearly explain the motivation behind converting time series data into images.
>
> One of the core innovations of this work lies in converting time series data into image representations to fully leverage the powerful feature extraction and transfer learning capabilities of visual models. Many works have explored the significant advantages of visualized time series data:
>
> - Images can reveal trends, cycles, anomalies and change structures in multivariable time series through spatial patterns. **These visual signals have higher recognisability for revealing the characteristics of time series information**[1].
> - Mant studies and reviews have pointed out that **visual models have rich pattern recognition capabilities in the image space**[2][3]. For instance, **convolutional networks and multimodal large models can automatically model local and global relationships in images**, which is particularly important for capturing multivariable dependencies and complex patterns in time series analysis.
> - **Image methods are conducive to multimodal fusion and cross-domain learning.** In time series scenarios with long lengths and many variables, they can efficiently integrate image output with text, numerical features, etc., achieving improvements in generalization and computational efficiency.
>
> Therefore, we aim to enrich the multi-angle information features of time series modeling through image modality, enhance the model's capabilities, and make it more robust. Meanwhile, multimodal fusion is a very promising direction, and our work can also provide an idea for the fusion of complex modal information.
>
> [1] Ni, J.,et al.Harnessing vision models for time series analysis: A survey. IJCAI2025
>
> [2] Wang, Y., et al.Utilizing image transforms and diffusion models for time series analysis. NeurIPS2024
>
> [3] Liu, Q.,et al.Time-VLM: Exploring multimodal vision-language models for unified time series forecasting. ICML2025
>
> > **W2**: It would be beneficial to include additional experiments to verify which features image-based methods are particularly good at extracting from time series data.
>
> Thank you for this valuable question. In our research, we systematically compared a variety of image-based feature extraction methods—including Recurrence Plot (RP), Gramian Angular Summation/Difference Field (GASF/GADF), and Markov Transition Field (MTF)—and referenced authoritative literature to determine the most suitable strategies for extracting time series features through imaging approaches (see Wang & Oates, 2015; Ni et al., 2025; Liu et al., 2025). Our experiments demonstrate that specific image transforms are particularly good at capturing distinct attributes:
>
> - Recurrence Plots are highly effective for extracting periodicity and recurrence patterns.
> - Gramian Angular Fields excel in encoding global temporal correlations and trend information.
> - Heatmap-based methods and Markov Transition Fields can reveal multivariate dependencies and dynamic state transitions.
>
> We plan to further enrich and expand these experimental results in future submissions, including more comprehensive evaluations across diverse time series benchmarks. These analyses will clarify which image-based methods are optimal for specific feature types and tasks. Thank you for highlighting this important area—we welcome continued dialogue and collaborative exploration.

---

> > ### Author Response · Authors · 2025-11-21
> > **Response to Reviewer 8Req [Part 2]**
> >
> > >**Q1:** A couple of confusion regarding channel dependency search: (1) I’m curious about why we specifically choose K channels. In multi-channel time series, the relationships between the channels are interconnected and serve different purposes. How can we determine that there is always one or several channels that are "most representative"?(2) Is the number of channels, K, treated as a hyperparameter in the paper? Considering that the relationships between channels can change dynamically with events, would it make sense for K to also change over time? And could the value of K vary depending on the specific dataset or domain?
> >
> > (1)**The selection of K representative channels**: Our approach leverages channel dependency search to identify a subset of channels that are most representative in terms of the overall structural coverage within the data.  Channels that are not selected are still incorporated via distance-weighted aggregation, ensuring minimal information loss.
> >
> > (2)**K is treated as a hyperparameter**: We performed a sensitivity analysis (Fig. 7) and found that K = 20 yields the best performance across multiple benchmarks. However, we recognize that K may vary depending on the **dataset and domain**, and it is advisable to tune K on the validation set in practice. The idea of allowing **K to change dynamically over time** is an interesting extension, and while it introduces additional complexity, it is a potential direction for future work.
> >
> > >**Q2**:  I wonder what the effect of shuffling channels would be on time images encoder. Since images have inherent spatial relationships, do the channels of the multivariate time series also carry spatial significance? Given that different datasets may have channels that represent different physical meanings, would it still be reasonable to convert them into time images to capture spatial information?
> >
> > We sincerely thank the reviewer for raising these thoughtful questions. They touch on important conceptual aspects of our imaging design, and we are glad to clarify them in detail.
> >
> > **(1) Effect of Shuffling the Channels**
> >
> > TimeCDS is specifically designed to be **insensitive to the permutation of raw channels**. The Channel Dependency Search (CDS) module builds a graph of inter-channel relationships based on similarity patterns, not on the original channel order. Channels are reordered according to their dependencies, and discarded channels are softly fused back using similarity-based aggregation. As a result, the final K-channel representation reflects **learned relational structures**, not the raw order of the channels. Therefore, shuffling the channels does not affect the encoding, as the dependency graph and the reconstructed representation remain stable.
> >
> > **(2) Whether Multivariate Channels Carry Spatial Significance**
> >
> > The spatial structure in TimeCDS is **computational, not physical**. While raw time-series channels do not possess physical spatial layouts, the relational matrix and periodicity–phase maps capture **intrinsic temporal structures** and **statistical dependencies**. Channels are reorganized based on statistical affinity, not physical proximity. The "spatial" layout in the resulting images corresponds to **learned relationships** such as correlation and temporal alignment, allowing the visual encoder to interpret local neighborhoods as statistically significant regions.
> >
> > **(3) Suitability Across Datasets with Different Channel Semantics**
> >
> > TimeCDS can be applied to **heterogeneous datasets** with different channel semantics (e.g., sensors, KPIs, climate variables). The imaging step leverages **universal time-series properties**, including temporal periodicity, cross-variable dependencies, and frequency-phase behaviors, which are consistent across domains. These properties define the **mathematical structure** of the signals, not their physical meaning, ensuring that TimeCDS generates valid and coherent representations regardless of the dataset's physical context.

---

> ### Author Response · Authors · 2025-11-21
> **Response to Reviewer 8Req [Part 3]**
>
> > **Q3**:  In Section 3.4.4, it is suggested to consider comparing TimeCDS with other image/CNN-based methods, such as the classic InceptionTime, Rocket, and others, to highlight the advantages of the proposed time image encoder.
>
> Based on your valuable suggestions, we have supplemented our analysis by including comparisons with several  methods, **InceptionTime, Rocket**,  and **TCN**. These comparisons provide a comprehensive evaluation of the performance of  our proposed time image encoder and help to demonstrate its unique strengths and contributions. The results are now reported as follows(evaluation metric: accuracy(%)):
>
> | Datasets / Models     | Rocket | InceptionTime | TCN  | TimeCDS |
> |-----------------------|--------|---------------|------|---------|
> | EthanolConcentration  | **44.8** | 31.6         | 28.9 | 42.7    |
> | FaceDetection         | **75.5** | 65.0         | 52.3 | 68.6    |
> | Handwriting           | **56.4** | 29.4         | 32.0 | 52.1    |
> | Heartbeat             | 74.7   | 75.6         | 76.1 | **78.6** |
> | JapaneseVowels        | 97.2   | **99.8**     | 98.7 | 99.8    |
> | PEMS-SF               | 73.6   | 81.3         | 81.6 | **85.8** |
> | SelfRegulationSCP1    | 90.0   | 90.0         | 89.3 | **91.2** |
> | SelfRegulationSCP2    | 52.4   | 53.7         | 53.4 | **57.2** |
> | SpokenArabicDigits    | 96.6   | **100**      | **100** | 99.0  |
> | UWaveGestureLibrary   | **92.1** | 85.6         | 85.4 | 86.7    |
> | Average Accuracy      | 75.33  | 71.2         | 69.77 | **76.17** |
> > **Q4**: Vectorization of Figures 1–3.
>
> We appreciate your important suggestions, which is very beneficial for the presentation of our paper. In the revised version Figures 1–3 have been updated to high-quality vector graphics to ensure improved clarity and scalability across  different viewing settings.

---

> > ### Comment · Reviewer_8Req · 2025-11-25
> >
> > Thank you for your detailed response. It basically solved my confusion. Regarding “W1: the motivation behind converting time series data into images”, I’d like to share a paper with you: “VisionTS: Visual Masked Autoencoders Are Free-Lunch Zero-Shot Time Series Forecasters.” It provides a clear explanation of why image-based methods can effectively capture the characteristics of time series data, and I hope it may offer you some inspiration.

---

> > > ### Author Response · Authors · 2025-11-26
> > > **Response to Reviewer 8Req**
> > >
> > > Dear Reviewer 8Req,
> > >
> > > Thank you very much for highlighting the motivation of converting time series data into images and for sharing the insightful VisionTS paper.
> > >
> > > As noted in VisionTS,  "Images can be regarded as 2D sequences of pixel values — exhibiting typical features of real-world time series such as trend, seasonality, and stationarity…" This observation underscores that both time series and images are natural signals with inherent structures, sharing similar underlying characteristics.
> > >
> > > Based on your valuable suggestions, we have revised the papers by incorporating the following clarifications in the second paragraph of Introduction.
> > >
> > > > Especially, VisionTS demonstrates that appropriate visual representation exhibits typical time series features, e.g., trend, seasonality, and stationarity, facilitating temporal dependency capturing.
> > >
> > > We will upload the fully-revised version of this paper soon by addressing all problems from the rebuttal period. Thank you again. We are wondering if our response addressed your concerns. If yes, we would be deeply grateful that you can kindly raise your score. We are more than happy to provide additional clarifications and discussions.
> > >
> > > Sincerely,
> > >
> > > Authors

---

### Official Review · Reviewer_7pQD · 2025-11-01

**Soundness:** 2
**Presentation:** 2
**Contribution:** 2
**Rating:** 4
**Confidence:** 3

**Summary:**

This paper proposes a channel dependency search module to model time series data under a unified scenario. The proposed framework handles different tasks at the same time and empirical results show its effectiveness.

**Strengths:**

1. Time series analysis is critical research field with solid motivation, especially under a multi-task scenario.
2. The proposed channel aware searching is reasonable to flexibly adapt into different tasks in a unified time series modeling framework.

**Weaknesses:**

1. Overall format needs significant improvement. Table is too small to read and figures are squeezed too much. They affect the readability of this draft.
2. It is hard to tell the proposed framework is novel enough, which is more like a combination of previous methods.
3. The term "search" of the proposed method is a little confusing. There is no searching operation, it is more like a graph learning concept.
4. That will be great if the comparison methods can be referred to corresponding papers in the experimental tables.

**Questions:**

Please check the above section.

---

> ### Author Response · Authors · 2025-11-21
> **Response to Reviewer  7pQD**
>
> Many thanks to Reviewer 7pQD for providing a valuable review. We have provided the following responses to the issues you are concerned about. We welcome you to continue discussing related matters with us.
>
> > **Q1**:  Overall format needs significant improvement. Table is too small to read and figures are squeezed too much. They affect the readability of this draft.
>
> We thank the reviewer for pointing this out. We will significantly improve the layout in the revised version, including enlarging tables, rescaling figures, and adjusting spacing to ensure better readability.
>
> >  **Q2**: It is hard to tell the proposed framework is novel enough, which is more like a combination of previous methods.
>
> Generally, the novelty of TimeCDS lies in the problem formulation and the mechanisms specifically designed for universal time-series modeling. In particular, we propose **Channel Dependency Search** as a solution to address cross-domain dimensional heterogeneity, thereby resolving a critical challenge that has not been effectively tackled by previous dual-branch or multimodal models. Second, **the time-series imaging decomposition** constructs domain-specific spatial structures rather than applying conventional vision modules. Last but not least, we propose an **adaptive cross-modal fusion,** which is designed for temporal–spatial complementary representations of the same signal, differing fundamentally from standard multimodal fusion.
>
> > **Q3**: Clarification of the term “search” in Channel Dependency Search.
>
> Thank you for highlighting the potential confusion. Our intention in using the term “search” is to emphasize that the representative channels are identified through **explicit similarity-search operations on the HNSW graph**, where nearest-neighbor retrieval determines the centrality and representativeness scores. The subsequent distance-weighted aggregation also depends on these search-derived neighborhoods.
> In the revision, we will more **clearly articulate the notion of “search” within CDS**, emphasizing that it refers to the explicit nearest-neighbor retrieval process on the constructed dependency graph.
>
> > **Q4**: Adding citations for comparison methods in tables.
>
> We thank the reviewer for the helpful suggestion. We will add references to all compared methods directly in the table captions or footnotes to enhance clarity and traceability.

---

### Official Review · Reviewer_MaBJ · 2025-11-02

**Soundness:** 2
**Presentation:** 2
**Contribution:** 2
**Rating:** 4
**Confidence:** 3

**Summary:**

The paper proposes TimeCDS, a universal, modality-aware framework for multi-task time-series analysis. It attempts to overcome the dimensional-heterogeneity problem across datasets by (i) selecting a fixed number of representative channels via HNSW-based channel-dependency search, (ii) converting the reduced series into an “image” representation, and (iii) fusing temporal and spatial features through a dual-branch Transformer/CNN encoder plus a cross-modal attention module. Extensive experiments on forecasting, anomaly detection and classification report consistent gains over ten or more baselines.

**Strengths:**

(1) Tackling heterogeneous variable counts and multiple tasks with one model is an important open problem.
(2) Channel-dependency search with soft distance-weighted fusion is new, and the imaging pipeline (periodicity + relation matrix + phase-amplitude) is creative.
(3) Best average MSE/MAE on 8 forecasting sets, highest F1/AUC/PATE on 5 anomaly sets, and top accuracy/F1 on 10 UEA classification sets; ablations show each module contributes.
(4) Well-organized structure, easy-to-follow notation, comprehensive appendix.

**Weaknesses:**

(1) Dual-branch Transformer+CNN with cross-attention resembles prior vision-language or multimodal models; novelty is mostly in the channel-search and imaging steps.
(2) No justification why 20 channels suffice, no analysis of information loss after discarding N−K channels.
(3) HNSW search + dual-branch forward pass + Cross-Modal Attention Mechanism (CMAM) is heavy; runtime/memory vs. baselines not reported.
(4) Foundation-model baselines (Timer, UniTS) were fine-tuned on individual tasks, whereas TimeCDS uses joint pre-training on UTSD-4G; comparison is therefore slightly favorable to TimeCDS.
(5) Only “w/o branch” and “w/o CMAM” tested; no study on patch size, imaging choices, or HNSW hyper-parameters.

**Questions:**

See Weaknesses.

---

> ### Author Response · Authors · 2025-11-21
> **Response to Reviewer MaBJ [PART 1]**
>
> Many thanks to Reviewer MaBJ for providing a detailed and in-depth review. Our responses to  your concerns are listed as follows.
>
> >  **Q1**: Dual-branch Transformer+CNN with cross-attention resembles prior vision-language or multimodal models; novelty is mostly in the channel-search and imaging steps.
>
> We appreciate the reviewer’s concern. TimeCDS is fundamentally different from prior vision-language or multimodal models, as it introduces dedicated mechanisms for universal time-series modeling.
>
> - **Channel Dependency Search (CDS)** resolves cross-domain dimensional heterogeneity by reconstructing a unified representation from selected channels, a capability absent in standard dual-branch architectures.
> - **Time-series imaging decomposition** constructs domain-specific spatial representations based on periodicity, relation-matrix, and phase–amplitude components, moving beyond generic visual encoding.
> - **Adaptive cross-modal fusion** integrates temporal and spatial features of the same signal using sample-wise attention and weighting, fundamentally differing from standard multimodal fusion schemes.
>
> In summary, these targeted mechanisms in dimensional alignment, representation, and fusion clearly differentiate TimeCDS at a methodological level.
>
> >  **Q2**: No justification why 20 channels suffice, no analysis of information loss after discarding N−K channels.
>
> We appreciate the  valuable observation.  The choice of K = 20 is not arbitrary but follows both empirical and methodological considerations.
>
> (1) Empirically, sensitivity experiments over K=[10, 15, 20, 25, 30, 35] ,Figure 7shows a clear rise–peak–decline trend, with performance consistently highest around K=20.
>
> (2) Methodologically, CDS does not discard the remaining channels but projects them onto the selected K channels through distance-weighted aggregation, effectively retaining their information.
>
> > **Q3**: HNSW search + dual-branch forward pass + Cross-Modal Attention Mechanism (CMAM) is heavy; runtime/memory vs. baselines not reported.
>
> Thank you for your active question. We also took this aspect into consideration during the experiment. To reduce costs, we referred to other excellent works and adopted the FlashAttention strategy in the attention mechanism. In the revised version, we reported the complexity of the model and provided the operational efficiency compared with the baseline.Specifically as follows:
>
> | Module | FLOPs | Parameters | Memory Footprint |
> | --- | --- | --- | --- |
> | Channel Dependency Search | O(NlogN+K²) | O(K²) | O(NT+NlogN+K²) |
> | Time Series Encoding Branch | O(BK(T+MLd)) | O(K²d+Kd²) | O(BK(T+ML+Md)) |
> | Time Image Encoding Branch | O(BK(T+LK+LD)) | O(FKLT+K²) | O(FKLT+K²+KD) |
> | Cross-Modal Alignment | O(BL(d²+K)) | O(Kd) | O(BL(d+L)) |
>
> In the complexity expression, N represents the total scale of the time series data, T is the length of a single time series, K is the number of representative features selected, B is the batch size, M is the number of blocks the time series is divided into, L is the length of each block, d is the feature dimension, and F is the size of the convolution kernel. D is related to the parameters of the projection layer. These symbols jointly describe the computational overhead of the model at different operation stages.
>
> We **included quantitive results of the computational cost** on high-dimensional time series (ECL: 321 variables) with a long lookback length (672), leading to the context of 200k+ time points.
>
> | Model    | Training Speed (s/iter) | Inference Speed (s/iter) | Params (M) | Performance (MSE) |
> |:---------|:------------------------|:-------------------------|:-----------|:------------------|
> | TimeCDS  | 0.2426                  | 0.0454                   | 16.73      | 0.151             |
> | PatchTST | 0.0625                  | 0.0079                   | 16.21      | 0.162             |
> | Timer-XL | 0.2157                  | 0.0447                   | 15.86      | 0.164             |
> | Timer    | 0.0458                  | 0.0079                   | 15.86      | 0.162             |
>
> Results are consistent with our theoretical analysis: Under the same hyperparameters, parameter counts are almost the same, while the multiplier of flops is less than the number of variables . We will also take it as essential work to improve the model efficiency of channel-dependent models.

---

> ### Author Response · Authors · 2025-11-21
> **Response to Reviewer MaBJ [PART 2]**
>
> > **Q4**: Foundation-model baselines (Timer, UniTS) were fine-tuned on individual tasks, whereas TimeCDS uses joint pre-training on UTSD-4G; comparison is therefore slightly favorable to TimeCDS.
>
> We thank the reviewer for raising this important point. We acknowledge that the current manuscript did not clearly  articulate the pre-training settings. In fact, **Timer and UniTS in our experiments were also pre-trained on UTSD-4G**,  following their official protocols,  and subsequently fine-tuned on each downstream task—identical to the procedure used for TimeCDS.  Since we hope to use the UTSD-4G dataset of better quality and compare it with the sota models timer and Timer-XL, we chose to **unify the pre-trained models used in the experimental setup to UTSD-4G**.
>
> >  **Q5**: Only “w/o branch” and “w/o CMAM” tested; no study on patch size, imaging choices, or HNSW hyper-parameters.
>
> Thank you for your concern about parameter sensitivity. In our original version, the **parameter K** most relevant to HNSW hyper-parameters has already been reported. To further demonstrate the validity of the model, we chose to evaluate the hyperparameter sensitivity of TimeCDS on the widely recognized **ERA 5-MS benchmark**. The main ones are the **patch size P** and the **Lookback Length L** during the inference process. Our research results show that the optimal patch size is usually close to the predicted length because it avoids multi-step error accumulation. Meanwhile, the research on Lookback Length L found that the optimal lookback length is not necessarily the same as the length used during training, indicating that the appropriate selection of information length is effective, and the reasoning stage can be compatible with different lengths and dimensions of cross-domain time series data.We reported the visualization of the results in the paper. The following table shows the original results.
> | **Lookback Length** | **MSE** |
> | --- | --- |
> | 96 | 0.1792 |
> | 384 | 0.1690 |
> | 768 | 0.1665 |
> | 1152 | 0.1662 |
> | 1536 | 0.1663 |
> | 1920 | 0.1665 |
> | 2304 | 0.1660 |
> | 2688 | 0.1671 |
> | 3072 | 0.1664 |
>
> | Patch Size | MSE |
> | --- | --- |
> | 24 | 0.17 |
> | 48 | 0.168 |
> | 96 | 0.165 |

---

### Author Response · Authors · 2025-11-27
**Kind Request for Discussion**

We sincerely thank all the reviewers for your insightful reviews and valuable comments, which are instructive for us to improve our paper.

We're pleased that the reviewers affirmed our paper our work for its "**novel channel-dependency search and creative imaging pipeline**" (Reviewer MaBJ, 7pQD),  "**meaningful inter-channel modeling and well-balanced dual-branch design**" (Reviewer 29P7, 8Req, MaBJ),  and "**superior multi-task performance across diverse benchmarks**" (Reviewer MaBJ, 8Req, 29P7).

The reviewers raised insightful and constructive concerns. We believe we have addressed all the problems by providing sufficient evidence and requesting results. Here is the summary of the major revisions:

- **Clarity of Method and Motivation (Reviewer 8Req, 29P7, MaBJ)**: We have enhanced the motivation for time-series-to-image conversion, demonstrated its effectiveness and necessity, and provided thorough justification for the key parameter K with robustness analysis.
- **Precision of Terminology and Innovation (Reviewer 7pQD, MaBJ)**: We have provided more explicit definitions of the "search" terminology in the text to clarify its specific meaning in our context, while further highlighting our core innovations in channel-aware architecture and cross-modal.
- **Comprehensiveness and Rigor of Experiments (Reviewer 29P7, MaBJ, 8Req)**: We have expanded baseline comparisons to include advanced CNN/image-based methods and foundation models, while adding computational complexity analysis and eliminating potential misleadings in baseline establishment.
- **Quality of Presentation and Readability (Reviewer 7pQD, 8Req)**: We have optimized all figures and tables to high-resolution vector formats, improved layout clarity, and added complete reference citations for all compared methods in experimental tables.

In this paper, we propose TimeCDS, **a universal modality-aware framework** for time series modeling,  which includes one strategy for handling heterogeneous dimensionalities (based on Channel Dependency Search) and one architecture for various time series tasks (leveraging dual-branch encoding and cross-modal fusion mechanisms). Insightfully, TimeCDS demonstrates superior performance across diverse time series tasks, **including forecasting, anomaly detection, and classification**.  These results highlight the effectiveness of TimeCDS in leveraging cross-modal features for universal time series analysis.

We have uploaded ***the revised version*** of our paper. Since half of the discussion period has elapsed, we sincerely look forward to your reevaluation and would be deeply grateful if you could consider a higher rating. Thank you very much for your time and consideration!

---

### Author Response · Authors · 2025-12-01
**Summary of Contributions Acknowledged by Reviewers and Discussion Phase**

Dear Area Chairs,

We would like to provide a brief summary of our contributions acknowledged by reviewers and further clarify the problems during the rebuttal period.

All reviewers highly acknowledged the significance and novelty of our work.  In particular, we are pleased that the reviewers have affirmed our work for its "**novel channel-dependency search and creative imaging pipeline**" (Reviewer MaBJ, 7pQD), "**meaningful inter-channel modeling and well-balanced dual-branch design**" (Reviewer 29P7, 8Req, MaBJ), and "**superior multi-task performance across diverse benchmarks**" (Reviewer MaBJ, 8Req, 29P7).

The reviewers' constructive suggestions have primarily centered on **further clarification of the methodological design**, **additional validation in experiments**, and **refinements in presentation**. We have thoroughly addressed each concern in a point-to-point manner on November 21, 2025.  Prior to the suspension of comments, we received further acknowledgment from Reviewer 29P7 and 8Req. Regrettably, neither Reviewer MaBJ nor 7pDQ provided any further response.

The corresponding discussions are summarized as follows:

- **Reviewer 29P7** confirmed that our response resolved their concerns and encouraged final submission to **upgrade their evaluation to a positive recommendation**.
- **Reviewer 8Req** acknowledged the resolution of their issues and constructively shared a relevant reference, which has been integrated into our revised manuscript.
- **Reviewer MaBJ** raised concerns regarding the motivation, clarity of terminology, and experimental rigor. In response, we have **comprehensively enhanced the elaboration of the method motivation** and conducted a **robustness analysis of the key parameter K**. The paper now also **clearly defines the relevant terminology** and **highlights the core innovations** in channel-aware architecture and cross-modal learning. The experimental section has been **expanded** with comparisons against advanced methods and foundation models, and supplemented with a **computational complexity analysis** to ensure a more rigorous evaluation.
- **Reviewer 7pQD** pointed out issues regarding term accuracy and the readability of figures. We have **clarified the specific meanings** of terms such as "search" within the context. Furthermore, all figures have been **optimized into high-definition vector format** with improved layouts, and **complete references for all compared methods** have been added to the experimental tables, **significantly enhancing the presentation quality and professionalism** of the paper.

To address key challenges in time series analysis, including **heterogeneous data dimensions**, **fragmented task-specific models** and **underutilized multimodal information**, we propose TimeCDS, **a universal modality-aware framework** that unifies dimensional alignment through Channel Dependency Search and supports multiple tasks via a dual-branch encoding and cross-modal fusion architecture. Extensive experiments confirm that TimeCDS **achieves superior performance** simultaneously on **forecasting, anomaly detection, and classification**, establishing it as an effective and general-purpose solution for diverse time series challenges.

We greatly appreciate the time and effort you devoted and sincerely hope that our work wins your support.

Best wishes,

Authors

---

### Note · Authors · 2026-01-20

I have read and agree with the venue's withdrawal policy on behalf of myself and my co-authors.